# 5-HT Receptors and Temperature Homeostasis

**DOI:** 10.3390/biom11121914

**Published:** 2021-12-20

**Authors:** Irina P. Voronova

**Affiliations:** Department of Thermophysiology, Scientific Research Institute of Neurosciences and Medicine, 630117 Novosibirsk, Russia

**Keywords:** body temperature, serotonin receptor, 5-HT_1A_, 5-HT_2_, 5-HT_3_, 5-HT_7_

## Abstract

The present review summarizes the data concerning the influence of serotonin (5-HT) receptors on body temperature in warm-blooded animals and on processes associated with its maintenance. This review includes the most important part of investigations from the first studies to the latest ones. The established results on the pharmacological activation of 5-HT_1A_, 5-HT_3,_ 5-HT_7_ and 5-HT_2_ receptor types are discussed. Such activation of the first 3 type of receptors causes a decrease in body temperature, whereas the 5-HT_2_ activation causes its increase. Physiological mechanisms leading to changes in body temperature as a result of 5-HT receptors’ activation are discussed. In case of 5-HT_1A_ receptor, they include an inhibition of shivering and non-shivering thermogenesis, as well simultaneous increase of peripheral blood flow, i.e., the processes of heat production and heat loss. The physiological processes mediated by 5-HT_2_ receptor are opposite to those of the 5-HT_1A_ receptor. Mechanisms of 5-HT_3_ and 5-HT_7_ receptor participation in these processes are yet to be studied in more detail. Some facts indicating that in natural conditions, without pharmacological impact, these 5-HT receptors are important links in the system of temperature homeostasis, are also discussed.

There is no process in an organism independent of the temperature, just as there is no process without heat dissipation. In his monograph, [1] Professor K.P. Ivanov presents astounding calculations by Calver and Prat (1956 cited in [1]): 1 g of human body at 37 °C per unit of time releases heat that is 10,000 times bigger than that released by 1 g of the Sun. “Harmonization” of heat flows escaping the body and coming from outside, ultimately led to the occurrence of an important evolutionary acquisition that allowed vertebrates to spread almost throughout the entire territory of our planet, it being the stability of body temperature (homeothermy). This stability is provided by a wonderful physiological system named the system of thermoregulation [2,3]. Being the youngest homeostatic system of the body in evolutionary terms, the system of thermoregulation, when functioning, uses practically all previously formed organs and tissues, including the mediator systems of the brain.

One of these brain mediator systems is serotonergic. In vertebrate phylogeny, it appears quite early, long before the appearance of homeothermy. In jawless fishes (*Cyclostomata)*, the neurons of the serotonergic system are already well developed [4,5].

Information about the effect of this neuronal system mediator, serotonin (5-HT), on body temperature of warm-blooded animals appeared soon after the discovery of its presence in the brain [6] and in particular in hypothalamus [7]. In 1957, H. Bachtold and A. Pletscher [8] published their data on an increase of rabbit body temperature after intravenous administration of 5-HT. Additionally, in 1961, N. Canal and A. Ornesi (cited in [9]) observed an increase in the body temperature of a rabbit after injecting 5-HT into the cerebral cisterna magna. In 1963, “Nature” published an article written by Feldberg W. and Myers R.D. [10], in which they suggested that body temperature of warm-blooded animals was maintained at a certain level due to “delicate balance in the release of adrenaline, noradrenaline, and 5-HT in the hypothalamus.” After this publication, for more than ten years, there was not a single year without several papers devoted to the effect of 5-HT on body temperature of warm-blooded animals published. Doses of the drug, methods of administration, experimental animals varied, and the results were different too. By the end of 1980s, several hypotheses were formulated that give preference to the one or the other effect of 5-HT on the processes of maintaining body temperature. Myers (1981) [11], summarizing his works and the works of his colleagues, maintained the opinion about the role of brain 5-HT mainly as a heat production factor. A different point of view was substantiated by other authors [12,13,14,15]. According to their opinion, brain 5-HT acts primarily as a heat loss factor. It has also been suggested that the activity of 5-HT structures of the brain determines the set-point of thermoregulation [16], as well as that brain 5-HT is a thermoregulatory reaction modulator, expanding the zone of thermoneutrality in warm-blooded animals [17]. An explanation for this huge amount of conflicting data was found later, precisely—after it was discovered that 5-HT has a number of receptors that sometimes have exactly the opposite effect on body functions.

As of today, 14 types and subtypes of 5-HT receptors have been identified. This is more than any known mediator has got [18,19]. Each of these receptors is characterized by its own amino acid sequence, chromosomal localization, and well-defined structural and functional characteristics. [18,19,20,21,22,23,24,25,26,27]. These 14 types of receptors are the molecular basis of the infinite variety of physiological effects that are observed in 5-HT [18,19,28,29]. The change in body temperature of warm-blooded organisms is also one of its physiological effects. 5-HT receptors that definitely influence the body temperature of a warm-blooded organism are the following: the 5-HT receptor of the 1A type (5-HT_1A_), the 5-HT receptor of the 2 type (5-HT_2_), the 5-HT receptor of the 3 type (5-HT_3_), and the 5-HT receptor of the 7 type (5-HT_7_) [18,19,27,30,31]. The present article focuses solely on these receptors and their role in body temperature maintenance of warm-blooded organisms.

## 1. 5-HT_1A_ Receptor

The 5-HT_1A_ receptor belongs to the superfamily of the G-protein coupled receptor (GPCR). It is encoded by an intronless gene located on chromosome 5 in humans, on chromosome 13 in mice, and on chromosome 2 in rats [32]. The promoter of the 5-HT_1A_ receptor gene (*Htr1a*) includes some repressor elements [33] that can regulate this gene expression. Moreover, there is evidence that *Htr1a* transcription can be regulated by gluco- and mineralocorticoids [34,35,36]. Protein product of this gene consists of 422 amino acids in human and rat and 421 amino acids in mice [32]. After translation, it forms seven transmembrane domains, extracellular N- and intracellular C-terminals. The latter, together with the 3rd intracellular loop, communicates with G-protein. The 5-HT_1A_ receptor is bound to the Gi protein, which, after receptor activation, inhibits adenylate cyclase, resulting in a decrease of cyclic adenosine monophosphate (cAMP) formation [37]. These receptors also indirectly open G-protein-gated inwardly rectifying potassium channels (GIRKs) leading to neuron hyperpolarization, and also inhibit the opening of voltage-gated calcium channels [22,26,27,37]. In addition, there is evidence that the 5-HT_1A_ receptor can use a variety of non-canonical pathways, such as the extracellular signal-regulated kinase (ERK) signaling pathway, signaling via small G proteins, and some others [26,37]. The way of signal transduction depends on many reasons, among which are the tissue- and region-specificity and also the ability of 5-HT_1A_ receptors to form homo- and heterodimers [26,27,38].

5-HT_1A_ receptors are widely spread in the organism, both in the peripheral and central nervous systems [19,27]. In the brain, they are found on the neurons of the cortical structures, hippocampus, septum, hypothalamus, midbrain raphe nuclei, and a number of other structures [39,40,41,42,43,44,45]. Investigation of their distribution by different methods gave similar results. Moreover, in midbrain raphe nuclei, 5-HT_1A_ receptors are also located on 5-HT neurons, i.e., they are autoreceptors (presynaptic or somatodendritic receptors as opposed to postsynaptic ones or heteroreceptors, i.e., the receptors located on non-5-HT neurons) and regulate the activity of the 5-HT system itself [39,40,41].

5-HT_1A_ receptors attract attention of researchers, since they are widespread in the brain, play an important role in the development of the nervous system and its plasticity, affect the functioning of the brain 5-HT system, i.e., are involved in its autoregulation, participate in regulation of many physiological functions, control various behavioral reactions, and are also associated with a number of mental diseases [46]. Moreover, their activation, as a rule, leads to the development of hypothermia in animals at warm ambient temperature.

Indeed, in response to the administration of 8-hydroxy-2-(di-n-propilamino)tetralin (8-OH-DPAT), a selective agonist of 5-HT_1A_ receptors, the body temperature of animals at the thermoneutral conditions decreases by about 0.5–3.5 °C (Table 1). Such a decrease is achieved, as a rule, in 10–20 min and does not depend on the route of drug administration. Exceptions are some genetic models, for example, rats specially selected for a reduced hypothermic response after administration of 8-OH-DPAT [45], rats with a high level of aggression [47], some strains of mice [48,49], as well as mice that are genetically modified to lack 5-HT_1A_ receptors [50,51,52,53].

Goodwin et al. in 1985 [54] have posed a quite natural question about the central or peripheral origin of the 8-OH-DPAT hypothermic effect. They compared the effects of peripheral (0.25–10 mg/kg s.c.) and central (3 µg i.c.v) drug administration. (According to the authors’ calculations, taking into account the weight of the animal, 3 µg corresponds to 100–120 µg/kg). They found that the central administration of a lower dose led to a faster and deeper drop in body temperature, and concluded that the effect was of central origin. Later, other researchers came to the same conclusion [56,81]: Bagdy and To [81] comparing the effects of central and peripheral administration of 8-OH-DPAT, Gudelsky et al. [56] using xylamidine- an antagonist of 5-HT receptors that does not penetrate the blood–brain barrier.

Goodwin et al. [54] in the same article made an assumption, supported later by a number of other authors, that the hypothermic effect of 5-HT_1A_ activation with 8-OH-DPAT “might serve as a simple physiological means of studying central serotonergic mechanisms in vivo” [54,55,61,65]. At present, this effect is generally recognized and is widely used both for assessing the functional activity of 5-HT_1A_ receptors [47,82] and the comparative characterization of newly created ligands to these receptors [50,83]. Development of gene engineering opened a new side using the hypothermic effect of 5-HT_1A_ activation: its absence confirms successful knock-out of *Htr1a* [52,70].

As 5-HT_1A_ receptors are located in different regions of the brain and also on 5-HT neurons (i.e., are post- and presynaptic), Hjort et al. [84] made the suggestion that pre- and postsynaptic receptors may differ in their properties. So, it is not surprising that in the scientific world, the question arose about the localization of 5-HT_1A_ receptors involved in the realization of the hypothermic effect that occurs after their activation. Thus, repeated attempts have been made to shed light on this problem (Table 2).

Depletion of the neuronal 5-HT pool by para-chlorophinylalanine (pCPA) [90], as well as the destruction of 5-HT neurons by neurotoxin 5,7-dihydroxytryptamine (5,7-DHT) were the tools most often used to address this issue. A reduction in hypothermic response after pCPA administration indicates a dependence upon endogenous 5-HT stores, presynaptic receptor localization, affecting 5-HT synthesis and/or release, is therefore inferred. Conversely, if the receptor is postsynaptic a reduction in mediator availability might be expected to result in an adaptive receptor upregulation, yielding a supersensitive response. In the experiments with such protocols, Hjorht [55] observed a clear hypothermic effect when using a low dose of 8-OH-DPAT (0.03 mg/kg), which was ineffective without prior administration of pCPA. According to these results, he concluded that postsynaptic 5-HT_1A_ receptors mediated the hypothermic response. The same conclusion was reached by Hudson et al. [59] as well as Matsuda et al. [85], who obtained similar results. Meller et al. [65], despite the use of multiple doses of both pCPA and 8-OH-DPAT, did not receive any changes in the hypothermic response and, subsequently, the answer to the question posed. Goodwin et al. [54,58] also pretreated animals with pCPA, but observed the disappearance of the 8-OH-DPAT hypothermic effect, from this they concluded that hypothermia is caused by presynaptic receptors. In studies of these authors an even more compelling argument in favor of presynaptic receptors was the disappearance of the 8-OH-DPAT effect after the destruction of 5-HT neurons by 5,7-DHT. Bill et al. [64], using the same methods, confirmed the findings of Goodwin et al. [54] in mice, and obtained opposite results in rats. As a possible reason for inconsistency in their results with the results of Goodwin et al. [58], the authors suggested the difference in doses of the drug used. Hillegaart [9] was also prone to the conclusion that the participation of presynaptic 5-HT_1A_ receptors in the realization of hypothermia. She injected 8-OH-DPAT directly into the region of 5-HT neurons localization (dorsal raphe nucleus, DRN) and observed a clear hypothermic effect, although she did not dare to completely exclude the role of post-synaptic receptors. Bagdy and To [81] compared the effects of different doses of 8-OH-DPAT injected i.c.v. with the results of Hillegaart [9] (see earlier) and concluded that receptors in the close vicinity of the lateral ventricle (i.e., post-synaptic) may play a predominant role in hypothermic response. An attempt to find out whether pre- or post-synaptic 5-HT_1A_ receptors are involved in the development of hypothermia in humans was undertaken by Blier et al. [86]. They offered to volunteers a tryptophan-free diet (to lower 5-HT levels in the brain), followed by buspirone (5-HT_1A_ receptor agonist). The researchers did not receive any differences between the control and experimental groups of volunteers both in the temperature response to 5-HT_1A_ receptors or in the endocrine responses mediated by post-synaptic 5-HT_1A_ receptors. From these results, the authors concluded that the hypothermic response after 5-HT_1A_ receptors’ activation in humans was also mediated by a post-synaptic mechanism. Knapp et al. [45] bred strains of rats different in sensitivity to 8-OH-DPAT. The selection was based on the severity of hypothermia in response to the administration of this drug. As a result, lines with high (HDS) and low (LDS) sensitivity were obtained. In rats of six to ten generations, the researchers conducted a radioligand study of 5-HT_1A_ receptors in the animal brains and tried to compare the number of receptor binding sites in different areas with the severity of hypothermia. Significant differences in binding were found in specific limbic cortical projection sites, with the HDS line having the greatest density of sites. Body temperature responses correlated significantly with [^3^H]8-OH-DPAT receptor binding only in a few areas of frontal cortex. Binding in many other brain regions, including anterior and posterior hypothalami (regions long associated with body temperature regulation) and raphe showed no significant differences among the lines. Thus, the attempt undertaken by Knapp et al. [45] did not provide a clear-cut answer.

Impressive results in deciding which pre- or post-synaptic receptors mediate the hypothermic effect of 8-OH-DPAT were obtained using genetic engineering methods [53,87]. The authors of these studies created mice lacking pre- or post-synaptic 5-HT_1A_ receptors. When investigating 5-HT_1A_ receptor activation, they found that *Htr1a* auto-KO mice displayed no detectable body temperature decrease in response to 8-OH-DPAT [53,87]. On the other hand, *Htr1a* hetero-KO mice displayed full hypothermic response to 8-OH-DPAT [87]. Based on the results of these authors, it can be argued that the hypothermic effect of 5-HT_1A_ receptor activation, at least in mice, is mediated by presynaptic receptors.

In the end of the discussion on the role of pre- and postsynaptic receptors in the hypothermic effect of 5-HT_1A_ receptors’ activation by 8-OH-DPAT, it is necessary to mention the results of Hjorth and Magnusson [41]. These authors convincingly showed that 8-OH-DPAT acts on presynaptic receptors located on the bodies of 5-HT neurons. In their experiments, after unilateral axonectomy of the ascending pathways from 5-HT neurons, systemic 8-OH-DPAT administration changed the synthesis and release of 5-HT in the region of terminals only at the intact side.

It would be also interesting to discuss the following, which physiological processes lead to hypothermia in a warm-blooded animal after 5-HT_1A_ receptors’ activation.

The hypothesis that the cause of a decrease in body temperature in an animal after 5-HT_1A_ receptors’ activation by 8-OH-DPAT is the heat loss processes was made by Hjorth in 1985 [55]. He referred to his own unpublished observations: body temperature did not decrease when experiments were carried out at an ambient temperature of 32 °C. In the same year, Goodwin et al. [54] published data on the development of the hypothermic effect at different ambient temperatures. The decrease of body temperature in an animal after 5-HT_1A_ receptors’ activation by 8-OH-DPAT was lesser in higher ambient temperature. The authors made a logical conclusion that the difficulty in heat dissipation prevented the development of the hypothermic effect of 5-HT_1A_ receptors’ activation. Therefore, it is the heat loss from the surface of the body that is the reason for the occurrence of hypothermia.

A more detailed study was carried out by Lin et al. [71]. They recorded not only body temperature being an integral parameter, but also oxygen (O_2_) consumption, by which they estimated the heat production of the organism, skin temperature, by which they assessed convective heat loss, as well as water vapor content in expired air, on the basis of which evaporative heat loss was calculated. The experiments were carried out at room temperature. 5-HT_1A_ receptors activation was performed by injecting 8-OH-DPAT s.c. (500 µg/kg) or directly into hypothalamus (0.5 µg). Despite the different route of drug administration, the results were similar: in animals, along with a decrease in body temperature, metabolic rate reduced, and heat loss from the body surface increased. Thus, the evidence was obtained that the processes of both heat loss and heat production were involved in the appearance of hypothermic response to 5-HT_1A_ receptors activation. In our experiments performed on mice CBA [80], we also observed the development of hypothermic response accompanied by a significant decrease in heat production (estimated by O_2_ consumption) after 5-HT_1A_ receptors’ activation (40nM of 8-OH-DPAT, i.c.v.), although an increase in heat loss (estimated by tail skin temperature) was non-significant.

Estimation of the heat loss by the measurement of the skin temperature in the areas that are considered to be responsible for this process (for example, ears and tail in rodents, ears in rabbits) [91,92,93,94,95,96,97] is a classic technique in physiology of thermoregulation. An increase in skin temperature indicates a vasodilation, a decrease—a blood vessel constriction. The speed of blood flow in given areas also changes. Its changes occur in parallel with constriction or dilation of blood vessels and are reflected in changes of skin temperature. Lin et al. [71] evaluated heat loss from the change in skin temperature on the rat paw. Direct measurements of blood flow in areas responsible for heat loss upon 5-HT_1A_ receptors activation were carried out by Blessing [98] in collaboration with Ootsuka [99,100,101]. In the experiments of these authors, blood flow in the rabbit pinna, initially reduced in response to cooling or pyrogen treatment, returned to normal values after 8-OH-DPAT administration. Administration of selective 5-HT_1A_ receptor antagonist N-[2-[4-(2-methoxyphenyl)piperazin-1-yl]ethyl]-N-pyridin-2-ylcyclohexanecarboxamide (WAY-100635) canceled the effect of 8-OH-DPAT. In the same experiments, Ootsuka and Blessing [99,100,101] discovered another interesting fact: the effect of 5-HT_1A_ receptors on the activity of sympathetic nerves mediating vascular tone [99]. It is known that exposure to cold is accompanied by the activation of the sympathoadrenal system [102,103,104]. 5-HT_1A_ receptors activation decreased the activity of vasomotor sympathetic nerves to about 5% of their initial level in warmth [99], and also significantly inhibited the discharge of these nerves during cooling [99,100,101]. At the same time, 8-OH-DPAT, administered peripherally, did not change nerves discharge evoked by electrical stimulation of preganglionic sympathetic axons in cervical sympathetic trunk. Based on these results, the authors concluded that the sympathoinhibitory action of this compound is realized within the central nervous system (CNS), and not peripherally. In other words, 5-HT_1A_ receptors that inhibit the activity of sympathetic vasomotor nerves are found in the CNS pathways normally activating cutaneous sympathetic vasomotor nerves in response to cold.

Heat production in the organism is carried out continuously by all organs and tissues. However in emergency cases, when it must be increased to maintain constant body temperature of a warm-blooded organism (for example, at exposure to cold), oxidative processes in tissues, and primarily in brown adipose tissue (BAT), increase. Moreover, special forms of muscle contractile activity arise: so-called muscle thermoregulatory tone and shivering [2,3,105,106,107,108,109]. BAT is a special type of adipose tissue. Its main function is considered to be heat production (non-shivering thermogenesis) [110,111,112]. A study of the effect of 5-HT_1A_ receptors on thermogenesis in BAT was undertaken by Ootsuka and Blessing [101]. In their experiments, the authors initiated thermogenesis by exposure to cold, and assessed it by the change in temperature of BAT. 5-HT_1A_ receptors activation by administration of 8-OH-DPAT inhibited thermogenesis in BAT. Morrison [113] investigated thermogenesis in BAT initiated by leptin administration and also found that activation of 5-HT_1A_ receptors inhibited thermogenesis. Studies on 5-HT_1A_ receptors’ role in modulating sympathetic influences on BAT thermogenesis have revealed their inhibitory role in these processes as well. A decrease in sympathetic impulses in BAT was recorded both with peripheral [101,114] and with central [114,115] administration of 8-OH-DPAT. Mota et al. [114] concluded that *rostral raphe pallidus* played an important role in the inhibition of BAT sympathetic nervous activity (SNA) by 5HT_1A_ receptor. They found that microinjections of WAY100635 into this area prevented changes caused by 8-OH-DPAT and this prevention took place both when 8-OH-DPAT was injected exactly into this area, and intravenously. Previously, the significance of this region as a key region in the onset of hypothermia caused by 5-HT_1A_ receptors activation was declared [77]. Further studies revealed that inhibition of 5-HT_1A_ receptors in warm environment by WAY100635 injected into the *rostral raphe pallidus* led to an increase in BAT SNA [114]. From this, the authors concluded that under a warm condition, “there is tonic activation of 5HT_1A_ receptors in the *rostral raphe pallidus* that contributes to the warm-evoked inhibition of BAT SNA and BAT thermogenesis” [114].

Another source of heat production, especially important under exposure to cold, is the so-called shivering thermogenesis—heat production in skeletal muscles by specific types of its activity—shivering and thermoregulatory tone [116,117,118,119]. As it turned out, 5-HT_1A_ receptors have an inhibitory effect on this source of heat production too. In the experiments of Berner et al. [120] at an ambient temperature of 15–20 °C (conditions capable of causing shivering), 8-OH-DPAT, injected into the *nucleus raphe magnus,* dose-dependently reduced the temperature of the preoptic region of hypothalamus, total metabolism assessed by O_2_ consumption, and shivering assessed by the electrical activity of the neck muscles. The inhibitory effect of 5-HT_1A_ receptors on shivering thermogenesis was also confirmed in other experiments: during dialysis of 8-OH-DPAT into the *paragigantocellularis lateralis* region, a medullary region lateral to the midline *raphe*’ that contains 5-HT neurons [121], and into the *medullary raphe*’ region [122].

The effects of 5-HT_1A_ receptors activation on thermoregulatory reactions of a warm-blooded organism are summarized in Table 3. It can be seen that 5-HT_1A_ receptor’s activation affects both heat production and heat loss processes, changes in which together lead to the well-known phenomenon—hypothermia of warm-blooded organisms under the activation of these receptors. The influence of 5-HT_1A_ receptors (at least on blood flow and thermogenesis in BAT) is realized via their effects on the activity of sympathetic nerves (see earlier).

After experiments of Goodwin et al. [54], the effects of 5-HT_1A_ receptor activation by 8-OH-DPAT are considered as central, regardless of the route of drug administration. The central administration of 5-HT_1A_ receptors ligands is used, as a rule, in order to reveal the role of specific regions of the CNS in the reactions mediated by these receptors. Madden and Morrison [124,125] drew attention to the 5-HT_1A_ receptor in the spinal cord. They discovered that 5-HT_1A_ receptors in the spinal intermediolateral nucleus could take place in regulation of BAT SNA and thermogenesis induced by this activity, but their effect was opposite to that of brain 5-HT_1A_ receptors. Thus, there is no doubt that more detailed studies of the role of 5-HT_1A_ receptors from different CNS areas and their share in regulation of physiological functions of the body are required.

It is worth emphasizing that all the above-described data show evidence only of the POTENTIAL role of 5-HT_1A_ receptors in thermoregulatory reactions, since all the results are the responses to pharmacological impact on receptors. However, it would be interesting to find out whether these receptors are involved in thermoregulatory reactions in natural conditions or in conditions nearest to the natural ones. Such an attempt was made [126]. As a model of temperature action, the model of long-term adaptation to cold (four to six weeks at a temperature of 4–6 °C) according to Hart [127] was chosen. This model is well studied from the point of view of thermoregulation. In particular, with its use, changes in the characteristics of neurons in midbrain [128] and hypothalamus [129,130] were revealed in animals adapted to cold compared to control ones. Since the neurons in the brain stem, as well as most thermosensitive neurons in hypothalamus, are able to respond to microinjections of 5-HT [131,132], i.e., have 5-HT receptors in their structure, it was hypothesized that these receptors could participate in changes of neurons characteristics after adaptation to cold. For the 5-HT_1A_ receptor, a good correspondence between the mRNA level of its gene and the number of its protein molecules (or sites of specific radioligand binding) in different parts of the brain was shown [19,27,42]. So, the change in the number of 5-HT_1A_ receptors can be judged with a high degree of probability by the change in the mRNA level of its gene. Based on these conclusions, it was decided to evaluate the effect of long-term exposure to cold on the 5-HT_1A_ receptor in the brain by the expression of its gene. However, changes in the 5-HT_1A_ receptor mRNA level in the studied brain regions (cortex, hippocampus, hypothalamus) after prolonged exposure to cold were not observed [126]. It is possible that changes could be found in other brain structures or using other models of temperature exposure.

However, some works supporting the role of the 5-HT_1A_ receptor in body temperature regulation in warm-blooded animals during natural body reactions without pharmacological influence can be cited.

This is indirectly evidenced by electrophysiological studies of brain neurons. Thus, it was found that the majority of *raphe obscurus* thermosensitive 5-HT neurons decrease the frequency of their discharges upon exposure to heat and increase it upon cooling (2/3 and 3/4 of neurons, respectively) [133]. Nason and Mason [134] in the medullary raphe also observed increased discharge of the cells, which they identified as 5-HT ones, in response to cold. It is possible that an increase in the discharge of 5-HT neurons in response to cold indicates a possible decrease in the activity of presynaptic 5-HT_1A_ receptors, which, as it is known (see earlier), contributes to the development of hypothermia. Conversely, a decrease in the discharge of 5-HT neurons in response to heating may indicate an increase in the activity of these receptors and the development of reactions that prevent overheating.

When studying 5-HT_1A_ receptor gene expression in ground squirrels at diverse stages of hibernation–waking cycle, a decrease in the mRNA level of this receptor in the midbrain during arousal of animals was found [135]. Arousal from hibernation is a very active process when body temperature of an animal rises in just a few hours from the values slightly higher than 0 °C to the value of normal body temperature in warm-blooded animals [136]. A decrease in the 5-HT_1A_ mRNA level (and, presumably, the amount of the receptor itself, the activation of which contributes to the development of hypothermia) upon arousal of animals from hibernation seems to be a quite reasonable physiological reaction.

Moreover, there is evidence that genetic modification of this receptor affects body temperature in mutant mice. Kusserow et al. [137] reported that in home cage body temperature in male mice with 5-HT_1A_ receptor gene overexpression is lower in comparison with wild type (WT) mice. In mice with *Htr1a*KO, stress-induced hyperthermia is higher than in mice of the corresponding WT: 2.5 and 1.5 °C, respectively [138,139]. The stressing factors in these experiments were the placement of animals in a new cage or the injection of saline. In both cases, i.e., regardless of the type of stress, temperature responses were similar: the increase in body temperature in KO mice was greater than in WT mice. Richardson-Jones [51] found, that in mice with high and low 5-HT_1A_ receptor levels, temperature response to a new cage was different: body temperature in mice with the low 5-HT_1A_ receptor level elevated more than in mice with the high 5-HT_1A_ receptor level. These results directly indicate that the absence (or deficiency) of one of the elements of the thermoregulatory system—the 5-HT_1A_ receptor—can lead to serious shifts in the maintenance of the body temperature balance. In this case, the absence of the 5-HT_1A_ receptor, the activation of which contributes to a decrease in body temperature, leads to the development of more pronounced hyperthermia. It is important to note that the difference in body temperature of mice described by Olivier et al. [138], Toth [139], and Richardson-Jones [51] arises under stress. In a calm state, when animals are in their home cage, without any external influences on them, there is no difference in body temperature. This example perfectly illustrates the necessity of choosing an adequate model for physiological experiment.

Thus, under the pharmacological activation of the 5-HT_1A_ receptor, reactions contributing to a decrease in heat production and an increase in heat loss develop. This leads to reduction in body temperature. These reactions are carried out with the involvement of the sympathetic nervous system. There are the facts indicating that in natural conditions, without pharmacological influence, the 5-HT_1A_ receptor is an important link in the system of temperature homeostasis. There is no data that could elucidate the possible changes in the receptors themselves during the processes of thermoregulation.

## 2. 5-HT_2_ Receptors

5-HT_2_ receptors are another type of 5-HT receptors that can influence the body temperature in warm-blooded organisms. It should be emphasized that all three receptor subtypes: 5-HT_2A_, 5-HT_2C_, and 5-HT_2B_—may participate in this impact. Although in relation to the latter there is no direct evidence that its activation changes body temperature of an animal, its participation in metabolic processes (see below) suggests their imminent appearance.

5-HT_2_ receptors, as well as most 5-HT receptors, are metabotropic, i.e., they carry out their function through the G-protein, with which they are coupled. Nevertheless these receptors differ significantly from the other 5-HT receptors by secondary messengers. Most metabotropic 5-HT receptors realize their action through cAMP, inhibiting its activity (such as 5-HT_1A_ receptors) or, conversely, activating it (such as 5-HT_7_ receptors). In 5-HT_2_ receptors, secondary messengers are mainly phospholipases: the “canonical” —phospholipase C (PLC) and “noncanonical”—phospholipase A2 (PLA2) and phospholipase D (PLD), as well as extracellular signal-regulated kinase (ERK), which can participate in signal transduction from other 5-HT receptors too, being also a “noncanonical” mediator for them [26]. In addition, 5-HT_2B_ receptors can transduce a signal by activating nitric-oxide synthases—constitutive and inducible [27].

5-HT_2_ receptors gene structures’ comparison made it possible to conclude that the most ancient 5-HT_2_ receptor subtype is 5-HT_2B_, from which 5-HT_2A_ and 5-HT_2C_ subtypes subsequently originated [140].

All three types of 5-HT_2_ receptors are widespread both in the brain and the periphery [26]. Earlier, it was thought that 5-HT_2B_ was absent in the brain [141] later its presence was discovered. In general, for 5-HT_2_ receptors, as well as for other 5-HT receptors, the distribution of specific receptor binding sites (i.e., protein molecules of 5-HT receptors), with rare exceptions, corresponds to the distribution of their mRNA. In the brain, 5-HT_2A_ (along with 5-HT_1A_) is widely distributed [26], especially in comparison with the number of other 5-HT receptors. However, judging by the mRNA level, 5-HT_2C_ is the most abundant 5-HT receptor in the brain [142]. Nevertheless, one should always take into account the local distribution of receptors in the brain regions. 5-HT_2A_ receptors have high density in forebrain: in cortex, hippocampus, basal ganglia. The highest level of 5-HT_2C_ receptor mRNA was found in choroid plexus, hippocampus, and subthalamic and lateral habenular nuclei [143,144,145], its rather high level was observed in median raphe, as well as in hypothalamus [26]. In hypothalamus, the *Htr2c* mRNA level is an order of magnitude higher than mRNA levels of other 5-HT receptors [146]. The presence of the 5-HT_2B_ receptor is also noted in many regions of the brain, including dorsal raphe [147], cortex, and dorsal hypothalamus [27].

5-HT_2A_ receptors’ gene expression regulation is carried out by numerous promoter sites such as cAMP-response element, glucocorticoid receptor binding site [148], and others. 5-HT_2B_ and 5-HT_2C_ receptors’ mRNA is capable of forming splicing variants [149,150]. In addition, the 5-HT_2C_ receptor gene is characterized by the presence of sites at which its mRNA can be edited [151,152,153,154]. To this date, among all known GPCRs, the 5-HT_2C_ receptor is the only one for which mRNA editing has been proven [27,150]. In total, there are 32 editing permutations in the 5HT_2C_ receptor transcript encoding 24 receptor proteins [27,150,155]. The edited 5-HT_2C_ receptor binds worse to the G-protein leading to a decrease in receptor activity. It is possible that RNA editing is a way of receptor activity regulation [27,150,151,152,153,154,155,156], although such regulation at the level of the whole organism requires strong physiological evidence.

Studies of mice with 5-HT_2_ receptors genes knockout indicate the participation of these receptors in the vital processes of the body. Thus, 5-HT_2B_ knockout is associated with severe cardiac abnormalities and embryonic and postnatal lethality [26]. 5-HT_2C_-knockout mice have decreased general metabolism and altered expression of some mitochondrial genes in comparison with WT mice [157].

Different ways for 5-HT_2_ receptors’ activity regulation have been described. It may be post-translational modifications—glycosylation (for 5HT_2A_ and 5HT_2C_) or phosphorylation (for 5HT_2C_), internalization (for 5HT_2A_, 5HT_2B_), as well as the ability to form homo- and heterodimers [27]. However, how these remarkable properties of the 5-HT_2_ receptor affect their ability to influence body temperature of a warm-blooded organism is not yet clear.

Amino acid sequences of these three receptors have a fairly high degree of homology: from 45-50% in general [19] to 80% in transmembrane domains [27]. Therefore, it is not surprising that the synthesized ligands often have similar affinities to these three types of receptors, and the problem of their selectivity is still relevant [27]. Table 4 presents the results of studies devoted to 5-HT_2_ receptors agonists and antagonists effects on body temperature in warm-blooded organisms. The results obtained in mice are separately given in the special table (Table 5), and they will be discussed separately.

One of the first studies of temperature effects of 5-HT, that focused specifically on the receptors of this neurotransmitter and where the “opposing roles of 5-HT_2_ and 5-HT_1A_ receptors” were really identified, was the work of Gudelsky et al. [56]. These authors, by using 5-HT_1A_ receptors agonist 8-OH-DPAT, 5-HT_2_ receptors agonist 2-chloro-6-piperazin-1-ylpyrazine (MK-212), and known antagonists of these receptors, proved that 5-HT_2_ receptors’ activation had the effect on rat body temperature as opposed to those caused by 5-HT_1A_ receptors activation: animals developed hyperthermia. The hyperthermic response to the activation of 5-HT_2_ receptors depended on the dose of administered agonist and decreased with the administration of antagonists. The maximal increase in body temperature in these experiments was approximately 1.1 °C.

Another important fact discovered by Gudelsky et al. [56] is the evidence that hyperthermia caused by 5-HT_2_ receptors activation, as well as hypothermia in response to 5-HT_1A_ receptors activation, is a phenomenon of central origin. The authors, using 5-HT receptor antagonist xylamidine, which did not penetrate the blood-brain barrier, showed that blockade of peripheral 5-HT receptors did not alter the hyperthermia induced by 5-HT_2_ receptors activation.

A number of further studies on the effect of 5-HT_2_ receptors agonists on body temperature in warm-blooded animals also confirmed their hyperthermic effect, the magnitude of which depended on the dose of the administered compound (Table 4).

It is difficult to say which subtypes of 5-HT_2_ receptors are involved in this hyperthermic effect, especially if the affinities of the ligand for 5-HT_2A_ and 5-HT_2C_ are close. For example, pKi (a one of characteristic features of drug-receptor interaction) for DOI and the human 5-HT_2A_ receptor is 7.4–9.2, for DOI and human 5-HT_2C_ receptor pKi equals to 7.2–8.6 [89]. However, some authors, for example, Mazzola-Pomietto et al., argued that the increase in body temperature under the influence of DOI was the result of 5-HT_2A_ receptors activation [164], and under the influence of mCPP it was the result of 5 -HT_2C_ receptors activation [166]. These authors substantiated their conclusions both by the influence of corresponding antagonists on the temperature effect, and by the emergence of behavioral reactions typical for the activation of the corresponding subtype of receptors in animals.

Zhang and Tao [171] raised the question of external temperature influence on the hyperthermic effect caused by 5-HT_2_ receptors activation. They found that at higher ambient temperatures, the hyperthermic effect was more pronounced. The authors suggested the explanation being not only an undoubted decrease in heat loss with an increase in ambient temperature, but also an increase in 5-HT_2A_ receptors affinity for DOI at elevated temperature. As an argument they cited the results of Dalpiaz et al. [181]. Latter authors, using the radioligand binding method, investigated the thermodynamic characteristics of three types of 5-HT receptors (5-HT_1A_, 5-HT_2_, 5-HT_3_) in samples of anterior cortex of the rat brain at various temperatures—from 0 to 35 °C—and found that 5-HT_2_ receptor affinity increased significantly with the rise of incubation temperature. Other receptors under study did not demonstrate similar dependency. However, it should be emphasized that Dalpiaz et al. [181] used [^3^H]-ketanserin (antagonist) and unlabeled 5-HT (agonist) to characterize 5-HT_2_ receptors. Later, it was found that 5-HT and ketanserin had different binding sites [19] and could not be used together in the investigations similar to those of Dalpiaz et al. [181]. In the experiments of Zhang and Tao [171], the compound began to act in animal organisms under the same temperature conditions (the initial body temperature of rats was the same). As a result of agonist action, body temperature of animals increased, but the maximal effect was 2.7 °C, so it is hardly worth talking about any thermodynamic changes in the affinity of the receptors. At the same time, the explanations based on the heat loss change under different temperature conditions are quite correct and physiological.

Gudelsky et al. [56] also carried out a targeted study of the effects of 5-HT_2_ receptor antagonists on body temperature. The dose-dependent hypothermic effect that developed in response to the ketanserin or ritanserin administration highlighted the active role of 5-HT_2_ receptors in body temperature maintenance in warm-blooded animals: not only activation of these receptors leads to a change in body temperature, but also their blockade. Changes in body temperature under activation or blockade of the receptors were of the opposite nature. Later, the hypothermic effect of 5-HT_2_ receptors blockade was noted by Mazzola-Pomietto et al. [164] in rats, and Morishima and Shibano [175] and Naumenko et al. [179] in mice. Nevertheless in most studies, no shifts in body temperature in response to 5-HT_2_ receptor antagonists administration were noted, although these substances reduced the hyperthermic response of agonists.

The results of Won and Lin [173] are somewhat apart from the others. These authors found the hyperthermic effect of ketanserin. This is not the only thing that surprises in this work. Researchers conducted experiments at different ambient temperatures using different doses of the drug. Surprisingly, a given dose of the drug produced the same temperature effect regardless of the ambient temperature. The authors explain this by different patterns of thermoregulatory responses at different temperature condition. Sometimes it was O_2_ consumption that changed in response to drug and ambient temperature, whereas in other cases it was skin temperature or evaporative heat loss, etc. Unexpectedly, the well-functioning thermoregulation system (its responses were documented) maintained the animal body temperature depending on the dose of drug administered regardless of the ambient temperature. It would be very interesting to identify the reasons for such an unexpected reaction.

5HT_2_ receptor influence data on body temperature in mice (Table 5) serve as evidence for absence of hyperthermic effect of 5-HT_2_ agonists in these animals, except the experiments of Buchborn et al. [178]. Moreover, the hypothermic effects of 5-HT_2_ receptor agonists were also described [174,176]. However, the absence of recorded temperature effects of 5-HT_2_ receptor agonists in mice is not the result of sensitivity lack of 5-HT_2_ receptors to agonists in these animals: behavioral reactions developing in response to agonists of these receptors are perfectly clear and well defined. Obviously, the peculiarities of heat loss in mice can explain the results.

In mice, the ratio of body surface to its volume is greater than in other species listed in Table 4, and, accordingly, heat transfer is higher [91,92,93,95]. Body temperature is an integral indicator of the balance of heat production and heat loss. Attention should also be paid to the temperature conditions of the experiments. Thus, Buchborn et al. [178] conducted their studies on anesthetized animals placed on a heat-pad. The authors discovered the hyperthermic effect of the agonist when they raised the temperature of the heat-pad from 37 to 41 °C: after the injection of the agonist, body temperature of an animal rose by one degree compared to the injection of the solvent. The experiments of Fox et al. [176], in which the strongest (−4 °C) hypothermic effect of 5-HT_2_ agonist was recorded, were carried out at 20 °C, that is lower than in other similar works. The authors also pointed out that the animals were placed in a plexiglass cylinder, and this could also significantly affect heat transfer. In this regard, it is useful to return to the classical study of Gudelsky et al. [56]. These researchers performed their study at 29 °C, when they worked with 5-HT_2_ receptor agonist MK-212 and at 23 °C when they worked with 5-HT_1A_ receptor agonist 8-OH-DPAT. This nuance gave them the possibility to emphasize the specificity of the receptors’ influences on body temperature in animals.

However the decrease in body temperature by 4 °C observed by Fox et al. [176] is a value too large to be explained simply by the effects of heat loss, especially since it was observed only at the highest dose of the drug. Perhaps this dose caused some behavioral effects that the investigators did not record, but which could significantly affect the heat loss. The fact that hypothermia was prevented by the administration of an antagonist can be explained by the action of an antagonist on behavioral reactions: the pretreatment with selective 5-HT_2A_ antagonist MDL 11,939 (used in experiments of Fox et al. [176]) prevented these behavioral effects, and together with them the hypothermic effect of a high dose of TCB-2. Another explanation of hypothermia induced by high dose of TCB-2 appears to be the effect of this compound (in such dose) on peripheral 5-HT_2A_ receptors. Sugimoto et al. [182] i.p. treated ICR mice with various doses of 5-HT and observed a pronounced hypothermic effect comparable to those described by Fox et al. [176]. By administering various 5-HT receptor antagonists, the authors have shown that this effect is mediated by peripheral 5-HT_2_ receptors [182].

Both an increase in heat production and a decrease in heat loss from body surface (Table 6) lead to hyperthermia after 5-HT_2_ receptors’ activation. An increase in total metabolism, assessed by O_2_ consumption, in response to intrahypothalamic administration of DOI in rats was firstly noted by Lin et al. [71]. In Zucker rats, Hayashi et al. [170] also observed an increase in total metabolism measured by O_2_ consumption and CO_2_ excretion after 5-HT_2C_ receptors’ activation. As for the source of heat production, to date, only the participation of BAT in these processes after 5-HT_2_ receptors activation has been experimentally proven. Ootsuka and Blessing [101] showed this by measuring the temperature of BAT after s.c. administration of DOI, Nakamura et al. [172]—after i.p. injection of 25B-NBOMe.

A decrease in heat loss from body surface after 5-HT_2_ receptors activation was reported by Lin et al. [71], who noted a decrease in paw skin temperature in rats. This conclusion was also confirmed by Nakamura et al. [172], who observed a decrease in tail skin temperature in animals under the same treatment. An irrefutable argument in favor of this conclusion is provided by the experiments of Blessing et al. [101,169,183], in which the authors directly measured peripheral blood flow and in which its decrease after 5-HT_2_ receptor activation was recorded.

An important finding of Blessing and Seaman [169] is the evidence that the decrease in peripheral blood flow following 5-HT_2_ receptors activation is mediated by the sympathetic nervous system. In their experiments, after unilateral sectioning of the cervical sympathetic trunk, the ipsilateral ear pinna decrease in blood flow in response to DOI was abolished or substantially reduced. So, as the authors emphasize, “vasoconsticting effect of DOI is mediated by a CNS action of the drug, not by a direct action on peripheral vessels” [169].

A different point of view on the origin of hyperthermia caused by 5-HT_2_ receptors activation is shared by Nakamura et al. [172]. As indicated earlier, after 5-HT_2_ receptor agonist 25B-NBOMe administration, they observed a hyperthermic reaction accompanied by peripheral vasoconstriction and an increase in BAT temperature. However, 5-HT_2_ receptor antagonist sarpogrelate administration significantly reduced the severity of vasoconstriction and prevented the development of hyperthermia. On the basis of these results, the authors concluded that peripheral 5-HT receptors significantly contributed to the observed response. Nakamura et al. [172] used sarpogrelate, suggesting that it does not cross the blood-brain barrier. However, there are different opinions on this in literature [184,185,186]. As for an increase in BAT thermogenesis, the authors associate this phenomenon with the influence of peripheral 5-HT. According to the views of Crane et al. [187] and Hansson et al. [188], 5-HT receptors are present on adipocytes and some of them (including 5-HT_2_) suppress lipolysis. Nakamura et al. [172] tested the hypothesis about the effect of peripheral 5-HT on BAT thermogenesis. At temperature of 29 °C (the temperature at which hyperthermia developed in the experiments of these authors), the blood level of 5-HT in experimental animals has been lowered. Investigators elevated it by i.p. injection of the mediator. Enhanced level of 5-HT indeed reduced the time during which the temperature of BAT was high. Thus, this study of Nakamura et al. [172], as well as the above-mentioned investigation of Sugimoto et al. [182], draw attention to 5-HT_2A_ receptors in the peripheral organs, which can also affect the processes of heat production and heat loss.

Ootsuka et al. [183] tried to find out the CNS location of 5-HT_2_ receptors that affect the activity of sympathetic nerves. Electrical stimulation of neurons in the *raphe/parapiramidal* region resulted in increased sympathetic nerve discharges and decreased peripheral blood flow measured in rabbits on the ear and in rats on the tail. 5-HT_2_ receptor antagonist ((trans-4-((3Z)3-[(2-Dimethylaminoethyl)oxyimino]-3-(2-fluorophenyl)-propen-1-yl)-phenol, hemifumarate (SR 46349B, 0.1 mg/kg, i.v.)) administration significantly reduced the degree of vasoconstriction in animals caused by electrical stimulation, confirming the presence of 5-HT_2A_ receptors in this region. Moreover, in this and in subsequent work [188], the investigators have shown the existence of spinal 5-HT_2A_ receptors that contribute to sympathetically induced cutaneous vasoconstriction regulated by *raphe/parapyramidal* neurons in brainstem.

Thus, it has been shown that pharmacological activation and blockade of 5-HT_2A_ receptors can cause opposite changes in body temperature in a warm-blooded organism. However, the question always arises whether these mechanisms work in any physiological models. The answer to this question is positive, at least for stress-induced hyperthermia. Ootsuka et al. [189] showed that an increase in BAT temperature in rats under restraint stress can be attenuated by selective blockade of 5-HT_2A_ receptors. Beig et al. [190] and Sinh and Ootsuka [191] also blocked hyperthermia in social stress models by 5-HT_2A_ receptor antagonist.

The involvement of 5-HT_2A_ receptors in thermoregulatory reactions in vivo has been shown [126] in the model of long-term adaptation to cold [127]. The description of the model and the rationale for its use for this purpose were given earlier. 5-HT_2A_ receptors gene expression was studied in cold-adapted and control animals in frontal cortex, hypothalamus, hippocampus, and midbrain regions. The adaptation resulted in an increase in the 5-HT_2A_ receptor mRNA level in hypothalamus and its decrease in frontal cortex. In light of the above-mentioned pharmacological data that 5-HT_2A_ receptors activation contributes to an increase in heat production and a decrease in heat loss, an elevation in the mRNA level of 5-HT_2A_ receptors in hypothalamus in animals adapted to cold is perceived as a quite expedient phenomenon, since under the influence of these receptors, the mechanisms leading to improved protection from cold are activated. It should be emphasized that the detected changes turned out to be specific for hypothalamus, which is obviously associated with its role as a center of thermoregulation. In hippocampus and midbrain, cold adaptation did not change 5-HT_2A_ receptor mRNA expression, while in cortex it led to changes opposite to those observed in hypothalamus: the amount of mRNA of this receptor in cortex significantly decreased. It is difficult to say to what particular physiological processes a decrease in 5-HT_2A_ receptor gene expression in cortex may be related, but it is interesting to note that the opposite changes—an increased number of serotonin 5-HT_2_ receptors in frontal cortex—were observed in rats specially selected for increased ability to the development of hypothermia in response to 8-OH-DPAT administration [45]. A decrease in the number of 5-HT_2_ receptor binding sites in cortex was also noted after chronic activation of these receptors [192] by their agonist. 5-HT_2_ activation under exposure to cold seems reasonable, and under prolonged exposure to cold (cold adaptation), 5-HT_2_ activation appears to be chronic. So, a decrease in the 5-HT_2A_ mRNA level in cortex found in these experiments is quite consistent with the results of Buckholtz et al., [192]. However, the physiological meaning of changes in 5-HT_2_ receptors in this brain region as a result of chronic cold impact is not yet clear.

In addition to the evidence listed above for the involvement of 5-HT_2_ receptors in thermoregulation, these receptors have many functions that are also associated with maintaining the temperature balance of the whole organism, but they have not yet been studied in this vein. First of all, it is the effect of these receptors on eating behavior. Changes in eating behavior after activation or inhibition of receptors are investigated as an independent behavioral response. However, there is no information regarding correlations between eating behavior and the temperature balance of the body under influence of 5-HT receptors. 5-HT_2B_ receptors exert the greatest influence on eating behavior, all other 5-HT receptors in comparison with them in this respect “act downstream”, as the authors of the last review figuratively put [27].

Another important aspect of the 5-HT_2_ receptors action is their involvement in the regulation of carbohydrate and fat metabolism [26,27,146,150,185,187,193,194,195,196,197,198]. 5-HT_2_ receptors, along with many other 5-HT receptors, are expressed in both white and brown adipose tissue, take part in the differentiation of adipocytes, and influence lipolysis and lipogenesis.

For some tissues, the 5-HT_2_ receptors’ effect on mitochondria, the “energy stations of cells”, has been shown. For example, in mice with a knockout of the 5-HT_2B_ receptor gene, both morphological changes in the mitochondria of cardiomyocytes and changes in the functioning of a number of mitochondrial enzymes were found [199]. Harmon et al. [200] showed the effect of 5-HT_2A_ and 5-HT_2C_ receptors on mitochondrial biogenesis—“an intricate process that drives coordinated transcription of both mitochondrial DNA and nuclear-encoded genes to increase cellular mitochondrial content” in the kidney. Nonogaki et al. [157] described changes in gene expression for uncoupling proteins (UCPs), important mitochondrial proteins [201], in white and brown adipose tissues, as well as in muscle and liver, in 5-HT_2C_ knockout mice.

Thus, a large amount of experimental material indicating the involvement of 5-HT_2_ receptors in the processes of heat production and heat loss, as well as their participation in a number of intracellular metabolic processes has been accumulated. It would be very desirable to receive experimental evidence of how intracellular processes activated (or inhibited) by 5-HT_2_ receptors affect the functioning of the organism as a whole. Such a combination of molecular and organismic levels of research would undoubtedly be very useful for all branches of biological science.

## 3. 5-HT_3_ Receptor

5-HT_3_ receptors are the only ones among numerous 5-HT receptors that are ion channels [19,26,27,202,203]. A brief history of their research can be found in a mini-review of Lummis [203]. This receptor is a membrane structure of five symmetrically located subunits surrounding the ion-conducting pore. Each subunit has N- and C-terminals located outside the cell, four transmembrane domains, and an intracellular loop between third and fourth transmembrane domains. The extracellular terminals are responsible for ligand binding and, therefore, are the sites of action for agonists and antagonists. The second transmembrane domain is involved in the formation of the canal pore, and its sites, together with some sites of the intracellular loop, control the movement of ions across the cell membrane [204,205,206,207].

The subunits that form the 5-HT_3_ receptor are not always uniform. To this date, in humans, for example, five different subunits have been found (5-HT_3A_-5-HT_3E_). Subunits are proteins that are structurally similar, formed not as a result of alternative splicing, but as a result of the expression of the corresponding genes. In humans, the genes for subunits A and B of the 5-HT_3_ receptor are located in the 11th chromosome, C, D, E—in the third [32]. In addition to this abundance of subunits, their splicing variants and various mutations affecting the ion permeability of this channel have also been described [27,207,208,209]. The leading role in 5-HT_3_ receptor formation, apparently, belongs to a subunit A. Thus, a functioning homomeric 5-HT_3_ receptor can be obtained only by combining these subunits, and for the functioning of a receptor “assembled” from various subunits, at least one must be A. However, the most common are believed to be ion channels consisting of A and B subunits [27,207,210,211,212]. The features of heteromeric 5-HT_3_ receptors formation apparently are tissue-specific [203,210]. The pharmacological properties of the homo- and heteromeric 5-HT_3_ receptor can differ significantly [210,212].

For 5-HT_3_ receptor subunits, chaperone proteins have been described. They are likely to promote correct folding, oligomerisation, post-translational modification, and/or export from endoplasmic reticulum [213]. Glycosylation is suggested as a post-translational modification of 5-HT_3_ receptor subunits: sites for this have been found on both A and B subunits [214].

The 5-HT_3_ receptors are localized both in the CNS and in the periphery. In the CNS, they are expressed in many areas [215,216]. However the highest levels of this receptor were mapped within the dorsal vagal complex (nucleus tractus solitarius, area postrema, and dorsal motor nucleus of the vagus nerve). Relative to the dorsal vagal complex 5-HT_3_ receptor expression in forebrain is low [22,216]. At the periphery, the 5-HT_3_ receptor was found in a wide range of tissues, including gastrointestinal tract, immune cells, heart, blood vessels, skin, adipocytes [196,207,211].

At the cellular level, activating the 5-HT_3_ receptor regulates the Ca^2+^, Na^+^, and K^+^ ions flow, that leads to the change of membrane potential and evidences the involvement of this receptor in the mechanisms of signal transduction [202,207,215,217]. At the organism level there are data concerning the participation of 5-HT_3_ receptors in the function of gastrointestinal tract, in modulation of pain, alcohol dependence, and some affective disorder [211].

“However, almost 50 years after the identification of the 5-HT_3_ receptor, we are now at the threshold of discovering the physiological role of 5-HT_3_ receptors in the CNS” (Pascal Chameau and Johannes A. van Hooft, [217]). One such poorly researched physiological roles of the 5-HT_3_ receptor is its participation in the regulation of body temperature.

There are only a few studies related to this problem (Table 7). Mazzola-Pomietto et al. [218] were among the first to investigate the effect of activation of serotonin 5-HT_3_ receptors on body temperature. They found a hyperthermic (slightly more than + 0.5 °C) effect of m-chlorophenylbiguanide (mCPBG) 5-HT_3_ receptor agonist after its i.p. administration on rats. Blockade of 5-HT_3_ receptors (i.p. administration of antagonists bemesetron—MDL72222—or ondansetron) did not affect the body temperature of animals, although MDL72222 reduced the hyperthermic effect of mCPBG. Ondansetron did not influence the hyperthermic effect of mCPBG. The authors could not give a clear answer to the question why blockade of receptors by different antagonists has different effects on the hyperthermic response caused by receptor activation, and reasonably concluded that additional studies of temperature effects of 5-HT_3_ receptors are required.

In the studies of Kandasamy [219], a huge difference was found between the effects of i.p. and i.c.v. administration of 5-HT_3_ receptor agonist 2-methyl-5-hydroxytryptamine maleate (2-Me-5-HT). For 45 min, the centrally administered agonist dose-dependently reduced the animal’s temperature, which then remained at this level for more than 2 h. I.p. administration of the drug had no effect. 5-HT_3_ receptor antagonists—ondansetron and tropisetron—regardless of the administration route, did not affect body temperature of an animal in the state of normothermia. In hypothermic animals, these antagonists acted differently depending on the type of hypothermia. Kandasamy [219] investigated 2 types of hypothermia: (1) caused by central 5-HT_3_ receptors activation by their agonist 2-Me-5-HT and (2) caused by gamma irradiation of an animal. Tropisetron, regardless of the administration route, dose-dependently reduced the severity of hypothermia in both models. Ondansetron, administered both i.p. and i.c.v., also dose-dependently reduced gamma-irradiation-induced hypothermia, but had no effect on central 5-HT_3_ receptor-induced hypothermia. What the difference between these models of hypothermia from the point of view of the processes of heat production and heat loss is, and why the actions of 5-HT_3_ receptor antagonists in these models are different is not yet clear. The ability of ondansetron (0.5 mg/kg i.p.) to reduce hypothermia has been reported by Ngampramuan et al. [221]. In their experiments, hypothermia was associated with motion sickness induced by rotating rats in their individual cages in the horizontal plane.

Experiments by Martin et al. [220] confirmed the results of Kandasamy [219] that there was no effect on animal body temperature by 5-HT_3_ receptor agonist 2-Me-5-HT administered i.p. Testing the hypothesis about the possible effect of blockade of 5-HT_3_ receptors (ondansetron, 0.1 mg/kg i.p.) on body temperature of an animal in a state of LPS-induced hyperthermia (LPS 50 µg/kg i.p.) did not reveal such an effect. Time-courses of body temperatures in LPS-treated animals with pretreatment by ondansetron or vehicle were the same.

Another attempt to identify the possible effect of 5-HT_3_ receptors on body temperature was made by Naumenko et al. [79]. Such authors like Mazzola-Pomietto et al. [218] worked with 5-HT_3_ receptors agonist mCPBG, but, unlike their predecessors, tested the effect of the drug both with peripheral (i.p.) and central (i.c.v.) administration. A powerful hypothermic effect of the i.c.v.-administered drug was found. The effect both in the magnitude of the drop in body temperature and in the duration clearly depended on the dose: mCPBG (160 nmol, i.c.v.) reduced the body temperature of an animal by almost 4 °C and hypothermia lasted up to 7 h, the effect of 40 nmol mCPBG (i.c.v.) was about ࢤ2 °C and lasted about 3 h. Central administration of 5-HT_3_ receptor antagonist ondansetron reduced the severity of hypothermia. Peripheral administration of mCPBG had no effect on body temperature, even at high doses. Based on this, the authors made a logical conclusion that mCPBG did not penetrate the blood–brain barrier. The authors tested their conclusion that brain 5-HT_3_ receptors activation leads to hypothermia on mice of six inbred strains. Hypothermia induced by brain 5-HT_3_ receptor activation developed in animals of all investigated strains. The temperature decrease (after 40nM mCPBG i.c.v.) ranged from ࢤ2.6 °C in AKR/J mice to ࢤ5.5 °C in CBA/Lac mice. Later, this conclusion was confirmed on mice of eight inbred strains [30]. Comparing the discovered hypothermic effect of brain 5-HT_3_ receptor activation with the known effect of 5-HT_1A_ receptor activation, the authors emphasize its depth and duration [79]: a drop in body temperature after 5-HT_1A_ activation is usually about 1.5 °C (Table 1), with the exception of rare cases of hypersensitivity to an agonist for these receptors, and lasts no more than one and a half hours. At the same time, hypothermia caused by 5-HT_3_ receptor stimulation was more than −2.5 °C and its duration was up to 6–7 h.

The discovery of such a powerful effect of central 5-HT_3_ receptors activation on animal body temperature has raised the question about its thermoregulatory mechanisms. As it was revealed [80], activation of central 5-HT_3_ receptors decreases heat production (O_2_ consumption and CO_2_ expiration) and increases heat loss (to estimate this parameter, the temperature of the tail skin was measured). In addition, it was found that the metabolism switched to a more economical work mode. This conclusion was made based on the analysis of the respiratory exchange ratio (RER). The RER in m-CPBG-treated animals decreased during the first 15 min after injection and then became stabile. RER decrease takes place during exposure to cold [222,223] and starvation [222]. This is important because lipids are the metabolic substrates that provide more economical consumption of energy resource per unit of calories supplied when these substrates are oxidized. Thus, the m-CPBG influence on metabolic processes consists not only in a decrease in total metabolism, but also in switching the metabolism to a more economical type of energy consumption [80].

Knockout of the 5-HT_3_ receptor gene leads to significant metabolic changes: in mice with this mutation (*Htr3a* KO), O_2_ consumption increases and fat metabolism is altered—they are resistant to obesity induced by high fat diet [196]. The fact that O_2_ consumption in *Htr3a* KO mice increases coincides with the data on a decrease in O_2_ consumption after 5-HT_3_ receptors’ activation [80].

Thus, all available data indicate that central 5-HT_3_ receptors activation leads to a decrease in body temperature of a warm-blooded organism [30,79,219], and this occurs due to a decrease in heat production and an increase in heat loss [80]. Data on 5-HT_3_ receptor blockade are ambiguous. As for peripheral 5-HT_3_ receptors, according to a large number of studies [79,219,220], theirs activation has no effect on body temperature. The results of Mazzola-Pomietto et al. [218] contradict the others. It can, of course, be assumed that there are interstrain differences between Wistar rats (the object of the study by Mazzola-Pomietto et al. [218]) and Sprague-Dawley ones (the object of the study by Kandasamy, as well as Martin et al. [219,220]). Without any doubt, this assumption requires experimental verification. The fact that the peripheral 5-HT_3_ receptor does not affect body temperature is also evidenced by the results of Sugimoto et al. [182]. These authors investigated the effects of peripherally administered 5-HT on rectal temperature and observed dose-dependent hypothermia in experimental animals. 5-HT_3_ receptor antagonist tropisetron did not attenuate this effect.

5-HT_3_ receptors of adipocytes are also the peripheral 5-HT_3_ receptors and their role in metabolic processes of these cells is investigated intensively [196]. Working with a culture of immortalized brown adipocytes (IBA), Oh et al. [196] found that activation (by mCPBG) and blockade (by ondansetron) of the 5-HT_3_ receptor cause opposite changes in adipocytes functioning (experiments were performed at the background of the β-3 adrenoreceptor stimulation, without stimulation of β-3 adrenoreceptors 5-HT_3_ receptor effects were absent). Ondansetron increased mRNA expression of thermogenic genes, such as *Ucp1* in IBA. Conversely, m-CPBG decreased the *Ucp1* mRNA level in IBA. O_2_ consumption of the BAT increased as a result of 5-HT_3_ receptor blocking. Ex vivo experiment using primary BAT from *Htr3a* KO mice showed similar results. Thermogenic gene expressions, including *Ucp1* expression, were higher in BAT of *Htr3a* KO mice. Moreover mitochondrial biogenesis increased in BAT of *Htr3a* KO mice. Thus, studies by Oh et al. [196] revealed an inhibitory effect of the 5-HT_3_ receptor on BAT thermogenesis. However it is still not clear what the proportion of thermogenesis induced (suppressed) by the blockade (activation) of the adipocytes 5-HT_3_ receptor in the total heat production of the body is.

Information about the influence of the 5-HT_3_ receptor on the other source of heat in an organism—on shivering thermogenesis—can be partly found in clinical investigations. As usual, the authors study the influence of 5-HT_3_ receptor antagonists on post-operative shivering (POS). The conclusions of meta-analysis of such data are almost the same year by year. “…More high-quality randomized controlled trials with larger sample size are still required to draw a definite conclusion about the preventive efficacy of 5-HT_3_ receptor antagonists on POS prevention in the future” [224]; “…The perioperative administration of 5-HT_3_ receptor antagonists may be an effective measure for the prevention of POS in patients undergoing spinal anesthesia. However, further studies investigating different types of surgeries are required” [225]. In general, these investigations evidence that 5-HT_3_ receptors influence shivering thermogenesis, but the quantification of this influence is still required. The data about the role of these receptors in the processes of body temperature regulation in natural conditions would be very desirable.

## 4. 5-HT_7_ Receptor

The 5-HT_7_ receptor was the last 5-HT receptor to be discovered. The feature of its discovery lies in the fact that it was not discovered as a target for a new pharmacological compound. It was identified from the screening of cDNA libraries by several independent research groups in 1993 [226,227,228,229]. The 5-HT_7_ receptor belongs to the super-family of GPCRs, is metabotropic, and as all 5-HT metabotropic receptors consists of seven transmembrane domains and extracellular N- and intracellular C-terminalis. In transmembrane domains, its homology with 5-HT_1_ receptors is close to 50%, with 5-HT_2_ receptors—close to 40%. As of today, 5-HT_7_ receptor genes are identified not only for human and most usual laboratory models—rats and mice, but also for many other species. The human 5-HT_7_ receptor gene (*HTR7*) is located on chromosome 10, mice *Htr*7—on chromosome 19. *HTR7* consists of four exons and codes a protein of 479 amino acids, rodent (mice, rats) *Htr7* consists of three exons and codes a protein of 448 amino acids [32]. For the 5-HT_7_ receptor, splicing variants are shown, but they do not exhibit major differences in their functional properties [27], except for the differential pattern of internalization displayed by the human 5-HT_7_ receptor (d) isoform [230,231]. Tissue–specific splicing differences between spleen and brain were found in humans [232]. The existence of the 5-HT_7_ receptor pseudogene has also been reported [27]. However, it is described for human and rhesus monkeys, not for rats or mice [233,234].

The 5-HT_7_ receptor can undergo post-translational modifications such as N-glycosylation and palmitoylation. However, if the functional significance of N-glycosylation has not yet been clarified, palmitoylation is important for switching intracellular signal transduction [27,38,230,231,235,236,237].

Signal transduction by the 5-HT_7_ receptor realizes through different pathways. This receptor is positively coupled to adenylate cyclase through stimulatory Gs proteins, and its activation results in an increase in cAMP. In addition, it interacts with G_12_ protein and has also “non-canonical” signaling pathways: ERK and small G-proteins [26,27]. It has been recently demonstrated that these receptors may form homo- and hetero-dimers (with the 5-HT_1A_ receptor). However, dimerization does not affect 5-HT_7_ receptor mediated signaling [27,236,238].

The presence of 5-HT_7_ receptors was shown in both the periphery tissues and in the CNS of various species by a wide variety of techniques. The CNS 5-HT_7_ receptors were found in the cortex, hypothalamus, thalamus, hippocampus, and also the spinal cord. 5-HT_7_ receptors are involved in the processes of memory and learning, modulation of circadian rhythms, sleep, pain perception, and others [27,37,230,231,234,235,237]. Participation in body temperature regulation is one of 5-HT_7_ receptors functions too.

The discovery of 5-HT_7_ receptor’s ability to influence body temperature, as the discovery of the receptor itself, was not traditional. Usually, the investigator begins the study of the receptor role in body temperature regulation from regarding the temperature changes induced by high selective agonists for this receptor. Exactly in this way, the influence of 5-HT_1A_, 5-HT_2_, and 5-HT_3_ receptors on body temperature of experimental animals was found. As for the 5-HT_7_ receptor, the investigation of its role in maintaining body temperature began from the selective antagonist that attenuated the reduction in body temperature induced by other pharmacological compounds. Hagan et al. [239] were the first to establish the association between the 5-HT_7_ receptor and 5-HT-induced hypothermia. They demonstrated that 3-[(2R)-2-[2-(4-methylpiperidin-1-yl)ethyl]pyrrolidine-1-sulfonyl]phenol (SB-269970), a high-selective antagonist for the 5-HT_7_ receptor, attenuated the reduction in body temperature induced by nonselective 5-HT receptor agonist 5-carboxamidotryptamine (5-CT) in guinea pigs. Later, the ability of 5-HT_7_ receptor antagonist to attenuate hypothermic effect of 5-CT was confirmed by other investigators [240,241]. In parallel, evidence of 5-HT_7_ receptor effect on body temperature was obtained in mice with a knockout of the gene for this receptor [75,241,242]. Mice lacking the *Htr7* gene (*Htr7*KO), in contrast to WT animals, did not lower their body temperature in response to 5-HT administration [242], low doses of 5-CT [241,242], as well as low-dose of 8-OH-DPAT [75] (Table 8).

It is noteworthy that the basal body temperatures in *Htr7*KO and WT animals are the same. The absence of a difference in body temperature between *Htr7*KO and WT animals is noted by most authors who worked with such animals [242,243,244]. This indicates either powerful compensatory processes occurring in the thermoregulation system, if its really important link has been removed, or weak significance of the 5-HT_7_ receptor in maintaining the body’s temperature balance. These data undoubtedly draw attention to the question of the 5-HT_7_ receptor role in thermoregulation, but so far only the integral parameter—body temperature—has come to the attention of researchers.

It is interesting to compare the results of Sugimoto et al. [182] and Hedlund et al. [242]. These authors used the property of peripherally administered 5-HT to induce hypothermia in experimental animals. They treated animals with the same dose of 5-HT and obtained a similar hypothermic effect. However, in the study of Sugimoto et al. [182], this effect was blocked by 5-HT_2_ receptor antagonists, and in the one by Hedlund et al. [242], its emergence in mutant animals was prevented by the absence of 5-HT_7_ receptors. Thus, the question of what is the proportion of 5-HT_2_ and 5-HT_7_ receptors in the hypothermic effect of peripherally administered 5-HT remains open. It is necessary to emphasize that a high dose of 5-CT (3 mg/kg, i.p.) in the experiments of Hedlund et al., [242] caused hypothermia even in animals lacking 5-HT_7_ receptors, which suggests that other 5-HT receptors also took part in these processes.

The hypothermic effect of 5-HT_1A_ receptors agonist 8-OH-DPAT, which have developed after its administration in a stable, dose-dependent manner and regardless of the route of administration (see above), was also tested for the possible involvement of 5-HT_7_ receptors. 8-OH-DPAT does indeed have an affinity to both of these receptors: its pKi = 8.4–9.4 (for 5-HT_1A_ human); 7.3–7.5 (for 5-HT_7_ rat); 6.3–7.6 (for 5-HT_7_ human); 6.6 (for 5-HT_7_ mouse) [89]. Studies of the 5-HT_7_ receptors’ role in the hypothermic effect of 8-OH-DPAT were carried out in mice (including those with *Htr7* knockout) and in rats (Table 9). As expected, 8-OH-DPAT dose-dependently decreased body temperature in experimental animals (and at high doses even in *Htr7*KO mice). The pre-administration of 5-HT_1A_ and 5-HT_7_ receptor antagonists, separately or together, with varying success, depending on their dose, weakened the hypothermic effect of 8-OH-DPAT. The general conclusion from these studies was that both receptors may be involved in the hypothermic effect of 8-OH-DPAT. Taking into account the data from recent years, when joint localization of these receptors, their ability to form heterodimers and mutual influence were shown, this conclusion does not seem strange, but indicates the need for a thorough study of these mutual influences and their role for the organism as a whole. Moreover, the data indicating that 8-OH-DPAT-induced hypothermia was a result only of 5-HT_1A_ receptor activation have already been obtained [30] (see below).

Selective agonists of 5-HT_7_ receptors were developed somewhat later than antagonists, and their properties are still being refined [246]. The effect of one of the 5-HT_7_ receptor agonists, *N*-(4-cyanophenylmethyl)-4-(2-diphenyl)-1-piperazinehexanamide (LP-211), on animal body temperature was investigated by Hedlund et al. [243]. The object of the study was mice—WT and *Htr7*KO. LP-211 was administered i.p. at doses of 3, 10, and 30 mg/kg. The dose of 3 mg/kg did not lead to a change in body temperature in animals of both genotypes, and 30 mg/kg reduced temperature in all animals, although in animals lacking 5-HT_7_ receptors this decrease was less significant both in magnitude and in duration. Selective 5-HT_7_ receptor antagonist 3-[(2R)-2-[2-(4-methylpiperidin-1-yl)ethyl]pyrrolidine-1-sulfonyl]phenol (SB-269970) blocked the LP-211 effect on body temperature, confirming that a decrease in temperature was the result of 5-HT_7_ receptors activation. WAY-100135 did not induce any changes in the LP-211 effect on body temperature confirming the absence of 5-HT_1A_ receptors participation in this hypothermia. Brenchat et al. [244] tried to reveal the temperature effect of other 5-HT_7_ receptor agonists: (2S)-N,N-dimethyl-5-(1,3,5-trimethylpyrazol-4-yl)-1,2,3,4-tetrahydronaphthalen-2-amine (AS-19), 2-(2-(dimethylamino)ethyl)- 4-(1,3,5-tri-methyl-1H-pyrazol-4-yl)phenol (E57421), dimethyl-*{*2-[3-(1,3,5-trimethyl-1H-pyrazol-4-yl)-phenyl]-ethyl*}*-amine dihydrochloride (E55888). The objects of the study were also WT and *Htr7*KO mice. Agonists were injected s.c. at the doses of 10 and 20 mg/kg. The compounds reduced body temperature in animals: AS-19 (20 mg/kg) in WT mice—by 3.8 °C, in *Htr7*KO—by 2.4 °C; E57421 (20 mg/kg)—by 1.1 °C in WT and by 2.3 °C in *Htr7*KO, that is under the influence of E57421, the temperature drop in WT was less than that in *Htr7*KO. E55888 (20 mg/kg) did not cause change in body temperature of animals. At a dose of 10 mg/kg, neither AS-19 nor E57421 had any effect on body temperature. The authors believed that the reason for hypothermia development under AS-19 and E57421 in animals without 5-HT_7_ receptors is the nonspecific ligand binding to other 5-HT receptors. Trying to explain the lack of temperature effect in E55888, the most selective 5-HT_7_ receptor agonist in this study, the authors suggested that 5-HT_7_ receptors activation alone is not enough to affect body temperature [244]. However, it seems that an explanation for this was found in the work by Naumenko et al. [30].

Naumenko et al. [30] worked with 5-HT_7_ receptors agonist 4-[2-(methylthio)phenyl]-*N*-(1,2,3,4-tetrahydro-1-naphthalenyl)-1-piperazinehexanamide (LP-44). To examine its effect on body temperature, they used not only a range of doses, but also different routes of their administration: i.c.v. and i.p.. I.c.v. administration of LP-44 produced a considerable dose-dependent hypothermic response. Pick effect of the maximum used dose (41.0 nmol) was observed between 15 and 30′ after injection and reached more than a 3 °C decrease. This effect was significantly attenuated by pretreatment with SB-269970 (16.1 fmol, i.c.v.). In contrast to i.c.v. administration, i.p. injection of LP44 in a wide range of doses (1.0, 2.0, and 10.0 mg/kg) failed to affect body temperature. Based on these findings, the authors concluded that central, rather than peripheral, 5-HT_7_ receptors are implicated in regulation of hypothermia. This result, presumably, explains why the highly selective 5-HT_7_ receptors agonist E55888, administered peripherally, did not affect body temperature of animals [244]. Although the review of DiPilato et al. [246] states that E55888 is brain penetrant, it also notes that its pharmacokinetic data have not been studied well enough. So, it may be assumed that the compound did not have enough time to get into the brain and realize its effect. Naumenko et al. [30] also showed that LP-44 does not affect 5-HT_1A_ receptors: 5-HT_1A_ receptors antagonist WAY-100635 did not attenuate LP-44-induced hypothermia. That is, LP-44-induced hypothermia is the effect of activating only 5-HT_7_ receptors. At the same time, blockade of 5-HT_7_ receptors with the selective antagonist SB 269970 produced no effect on 8-OH-DPAT-induced hypothermia, nor on mCPBG (5-HT_3_ receptors)-induced hypothermia. In general, the obtained results allowed the authors to formulate the following conclusions: “1) 8-OH-DPAT specifically binds with the 5-HT_1A_ receptor; 2) the central 5-HT_7_ receptor plays an essential role in the mechanism of thermoregulation independent of 5-HT_1A_ and 5-HT_3_ receptors” [30].

It seems necessary to take into consideration and discuss in more detail one more fact discovered by Naumenko et al. [30], which, perhaps, was not discussed well enough in the work. The hypothermic effect of centrally administered 5-HT was not altered by the 5-HT_7_ receptor blockade. This indicates that for the natural ligand (albeit in the situation of an artificial increase of its concentration in the brain), the 5-HT_7_ receptor is not a “first line of response” receptor regarding body temperature changes. However, this fact can be also explained by the too high dose of 5-HT used (61.7 nmol). Although the authors of the study wrote that they had chosen the dose very carefully, in the experiments of Yamada et al. [247], the more pronounced temperature effect developed in response to the 5-HT dose that was six times smaller (10 nmol, i.c.v.). So, these results seem to demand further investigation.

Information on how exactly 5-HT_7_ receptors are involved in thermoregulation, whose processes of heat production and/or heat loss they affect, is given so far only in two works: Madden and Morrison [124,125]. These authors investigated the role of the intermediolateral (IML) nucleus (column) of the spinal cord in sympathetic regulation of BAT. They found that the blockade of 5-HT_7_ receptors in this region (microinjections of SB 269970) partially reduces BAT SNA, induced by some pharmacological drugs [124] or cold [125]. These results indicate that 5-HT_7_ receptors in the spinal cord contribute to increases in BAT SNA and thermogenesis, i.e., they act oppositely to 5-HT_7_ receptors in the brain.

Certainly the question arises: does this ability of 5-HT_7_ receptors to lower the body temperature of warm-blooded animals found in pharmacological studies have any physiological significance? Are there any facts confirming the participation of 5-HT_7_ receptors in maintaining body temperature in natural conditions? At the moment, unfortunately, there are very few studies shedding light on these questions.

In one of them, Kose et al. [248] tried to elucidate the role of 5-HT_7_ receptors in maintaining body temperature during LPS-induced hyperthermia in mice. It was found that 5-HT_7_ receptor agonist AS19 not only prevented the development of LPS-induced hyperthermia, but also led to a decrease in body temperature in experimental animals as compared to control ones (treated only by vehicle). 5-HT_7_ receptor antagonist (SB269970) promoted more pronounced hyperthermia even compared to LPS-treated animals. In other words, 5-HT_7_ receptors activation prevented the development of fever, and their blockade aggravated the fever. In general, based on known data on the effect of this receptor type on body temperature, such a result is expected. However, perhaps the most interesting fact shown in this work is that LPS increased *Htr7* expression in hypothalamus compared to the level of control animals. Correlation between the level of animal body temperature and the *Htr7* expression level was not found. This suggests that body temperature is not a trigger for increasing of *Htr7* expression in this study. However, the fact of an expression increase indicates active involvement of the 5-HT_7_ receptor in organism response to the bacterial toxin. Whether the task of the 5-HT_7_ receptor in response to LPS was: to reduce body temperature elevation or to mediate immune reaction is still unclear. Nevertheless this is an example of an active reaction of the receptor to external nonpharmacological influence.

Gargaglioni et al. [249] investigated the possible role of 5-HT_7_ receptors in the development of hypoxic hypothermia. It is known that hypothermia, accompanied by a decrease in metabolism, is an adaptive response used by many species of animals, especially small ones, to experience adverse conditions. Hypoxic hypothermia is a physiological response to the reduction of O_2_ in inhaled air. The authors induced hypothermia in experimental animals (rats) by placing them in an atmosphere with low O_2_ content. Blockade of 5-HT_7_ receptors by microinjection of SB269970 into anteroventral preoptic region (AVPO) caused a dose-dependent reduction in hypoxia-induced hypothermia. Microinjections of SB269970 into extra-AVPO region had no effects. So, the authors demonstrated not only the participation of 5-HT_7_ receptors in the development of hypoxic hypothermia, but also the precise region of the brain responsible for it.

Comparison of the known facts about the localization and physiological functions of 5-HT_7_ receptors, such as their high concentration in the supraoptic nucleus of hypothalamus, participation in regulation of circadian rhythms, sleep, changes in body temperature, suggests a possible role of these receptors in such adaptive reactions as daily torpor and hibernation. Hrvatin et al. [250] name torpor and hibernation “the most fascinating adaptations of warm-blooded animals, endowing them with the ability to survive in harsh environments otherwise incompatible with life”. Nevertheless, studies on the role of 5-HT_7_ receptors in these amazing processes are still awaiting their authors.

## 5. General Discussion and Conclusions

Speaking about the role of 5-HT receptors in thermoregulation, one should remember that the thermoregulation system, in addition to effector links, also has an afferent one, i.e., those through which the temperature information reaches the center. Unfortunately, studies on the effect of 5-HT receptors on this link of the thermoregulation system seem to be absent or unavailable. However, the possibility of such influence exists. It is confirmed by the effect of 5-HT receptors on pain sensitivity, including temperature pain sensitivity [244,251,252,253,254,255]. In addition, it was shown that microiontophoretic application of 5-HT suppresses the response of dorsal horn neurons to thermal skin stimulation [256]. This indicates a possible involvement of 5-HT receptors in the modulation of the temperature afferent signal. It is not yet clear what types of 5-HT receptors are involved in this process, but the possibility of their influence on temperature sensitivity is beyond doubt.

Certainly the intriguing question is the question of the mutual influence (interaction) of 5-HT receptors in the maintenance of body temperature. However, the answer on this question is not easy. Firstly, due to the fact that the results of receptor interactions at the cellular level studies (see [38,257]) or at the level of hypothalamic nuclei ones [258,259], as a rule, do not include studies of temperature effects. Secondly, when only temperature effects are investigated, the results of the actions practically represent the sum of opposite (or unidirectional) influences. So, for example, under the influence of 5-HT_2_ receptor activation, 5-HT_1A_-induced hypothermia is less pronounced, and under the 5 -HT_2_ receptor blockade—enhanced [56,63,179] and *vice versa*: hyperthermia caused by 5-HT_2_ receptors activation decreases upon 5-HT_1A_ receptors activation, and upon blockade of these receptors—increases [168]. At what stage is the “summation” of receptor effects—already at the level of the effector or earlier—at the level of central interactions, is difficult to say. The fact that, under the influence of some 5-HT receptors, behavioral reactions mediated by other receptors also change [49,179], allows to conclude that the interaction of receptors occurs at the center level. However, it would be desirable that changes in the characteristics of one receptor under the influence of another, significant for thermoregulation, but found in the study of behavior, were confirmed in appropriate experiments.

Thus, a lot of data has been collected concerning the effect of 5-HT receptors on body temperature of warm-blooded animals and on processes associated with its maintenance. It has been established that the pharmacological activation of 5-HT_1A_, 5-HT_3_, and 5-HT_7_ types causes a decrease in body temperature, whereas 5-HT_2_ activation causes its increase. The following fact is rather interesting. Thermoregulatory reactions aimed at increasing body temperature of a warm-blooded animal are controlled only by 5-HT_2_ receptors, which are similar in structure and signal transmission, while a decrease is controlled by three types of receptors that differ significantly not only in intracellular signal transmission (like 5-HT_1A_ and 5-HT_7_), but also are the compounds of different classes: GPCRs (5-HT_1A_ and 5-HT_7_) and the ion channel (5-HT_3_). Duplication of control over some parameter both in technology and in physiology testifies to the significance of this parameter. The fact that 5-HT receptors of three different types are “targeted” to reduce body temperature of a warm-blooded organism, emphasizes the fragility of the state of homeothermy and its balancing near the dangerous limit of overheating. From death due to hypothermia, warm-blooded animals (even humans) are separated by more than 15 °C, and from death due to overheating by just over 5 °C [1]. It seems that the 5-HT system with its three types of receptors, the activation of which contributes to a decrease in body temperature, stands on the protection of warm-blooded organism just from overheating. It would be useful to illustrate this suggestion. The results of Ishiwata et al. [260] may serve as a good illustration. These authors simultaneously recorded the content of monoamines (norepinephrine, dopamine and 5-HT) in the ventral tegmental area, body temperature (integral parameter), and tail temperature (for estimation of heat loss intensity) in non-anesthetized rats. The animals, after measuring these parameters at room temperature (23 °C), were subjected to 2 h of temperature exposure: heat (35 °C) or cold (5 °C). Under exposure to heat, an increase in body temperature (≈+1.5 °C), an increase in tail temperature (≈+10 °C), and an increase in the 5-HT level (by more than 30%) were recorded. There were no changes in the concentration of other monoamines. These results clearly confirm that it is 5-HT system that responds to exposure to heat. The next step was to find out the physiological meaning of the observed change. For this purpose, the authors modeled changes in the concentration of 5-HT in the studied brain area by injecting an inhibitor of its reuptake. An increase in the concentration of 5-HT was accompanied by a decrease in body temperature of a rat due to the activation of heat loss processes (an increase in the temperature of the tail). This is very similar to the processes occurring when 5-HT_1A_ and 5-HT_3_ receptors are activated. Exposure to cold in these experiments turned out to be more gentle, although its duration was the same as of one to heat. Body temperature of animals even increased, apparently, due to heat loss limitation, judging by the temperature of the tail, the concentration of monoamines did not change. That is, an increase in the ambient temperature by 12 °C had a more powerful effect on the rat’s organism than a decrease by 18 °C. A similar pattern is true not only for a rat, but also for other warm-blooded animals [91,92,93]. So the presence of three types of 5-HT receptors, the activation of which is aimed at lowering body temperature, is well justified.

Knowledge of the subtle mechanisms of influence on body temperature, which are inherent in the organism by nature, opens up prospects for a sparing effect on the body to obtain a state of hypo- or hyperthermia, if it is necessary (for example, an attempt to obtain a state of hypothermia using only external cooling invariably encounters intense resistance from the thermoregulatory system, see Burton and Edholm [261]). The areas where this knowledge can be used are the widest. It may be a decrease in postoperative tremors, and the possibility of influencing inflammatory processes, provoking hyperthermia to combat certain cancers or viruses and *vice versa*, introducing an injured patient into hypothermia to prevent damage to vital organs (brain, heart) during its transportation to clinics, prevention of radiation damage and, of course, the use of hypometabolic states in the conquest of interplanetary space [262,263,264,265].

In conclusion, it should be noted that until now, among numerous excellent reviews describing 5-HT receptors, only one (31) was devoted specially to the role of 5-HT receptors in body temperature maintenance. The ability of these receptors to change body temperature in warm-blooded animals was usually only mentioned in previous reviews, but not discussed. Present manuscript fills this gap; however, the task of any review is not only to summarize the known material (Figure 1), but also to indicate what has not yet been done. Changes in temperature homeostasis parameters under pharmacological influences have been studied to the greatest extent. The interpretation of the results obtained with the help of genetic modifications must always be approached with caution, taking into account possible compensatory processes in the body. Moreover, experimental evidence is needed for the participation of 5-HT receptors in thermoregulatory reactions under natural (or as close to natural as possible) conditions.

## Figures and Tables

**Figure 1 biomolecules-11-01914-f001:**
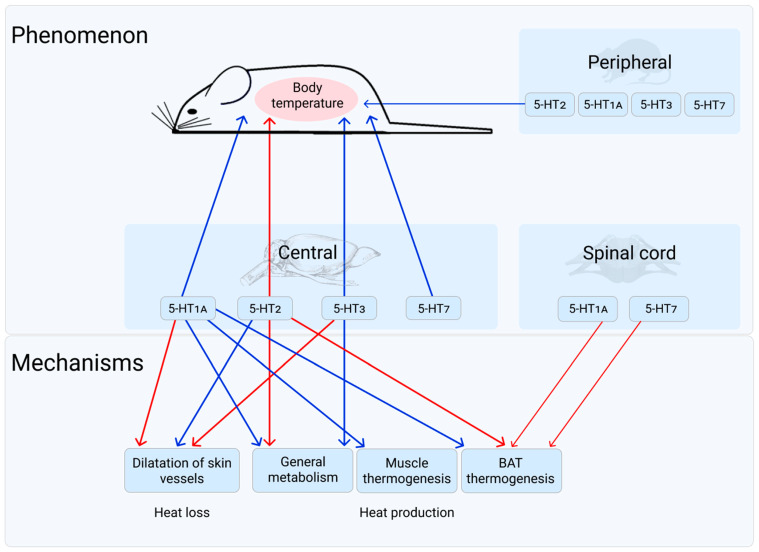
Phenomenon—changes in body temperature of warm blooded organism—and mechanisms of its appearance—physiological processes of heat loss and heat production that stimulation (inhibition) under 5-HT receptors activation lead to observed phenomenon. Red arrows—stimulation of physiological processes and increase of body temperature. Blue arrows—inhibition of physiological processes and decrease of body temperature.

**Table 1 biomolecules-11-01914-t001:** Maximal hypothermic effect of selective 5-HT_1A_ receptor agonist 8-OH-DPAT in warm-blooded animals.

Maximal Hypothermic Effect (°C)	Time of Its Achievements (min)	Object	Dose of 8-OH-DPAT	Route of Administration	References
3.5	10	C57/B16 mice	3.0 µg	i.c.v.	[54]
2.2	30		0.5 mg/kg	s.c.	
2.9	30		2.5 mg/kg		
3	18		5.0 mg/kg		
3	18		10.0 mg/kg		
2.4	30	Sprague-Dawley rats	0.03–0.1 mg/kg	s.c.	[55]
3	30	Sprague-Dawley rats	0.3 mg/kg	s.c.	[56]
3	15	C57Bl6 mice	5 mg/kg	s.c.	[57]
3.5	10		2 µg	i.c.v.	
2	30	Sprague-Dawley rats	0.3 mg/kg	s.c.	[58]
2.5	30	Sprague-Dawley rats	1 mg/kg	s.c.	[59]
2.5	10	Hooded Lister rats	2500 ng	into DRN	[60]
0.5	60	Wistar rats	0.125 mg/kg	s.c.	[61]
0.9	30		0.25 mg/kg		
1.5	60		0.5 mg/kg		
1.3	30	Sprague-Dawley rats	0.1 mg/kg	s.c.	[62]
1.6	10	ICR mice	0.25mg/kg	s.c.	[63]
3.0	15–30	Female albino Tuck (T/O strain) mice	3.0 mg/kg	s.c.	[64]
2.3	30	Sprague-Dawley rats			
1.3	20	Wistar rats	5 µg	into DRN	[9]
no effect				into MRN	
2.5	15	Swiss-Webster mice	2 mg/kg	s.c.	[65]
2	20	C57/Bl/6Ola mice	5.0 mg/kg	s.c.	[66]
1.3			0.5 mg/kg		
2.5	30	Wistar rats	0.5 mg/kg	s.c.	[67]
2	15–30	Sprague-Dawley rats	0.25 mg/kg	s.c.	[68]
2.8	15–30	T/O mice female	0.5 mg/kg		
1.5	30	Wistar rats	0.3 mg/kg	s.c.	[69]
4	20	C57/Bl6*Htr1a* (+/+)	1 mg/kg	s.c.	[70]
3.3		C57/Bl6 *Htr1a* (−/+)			
no effect		C57/Bl6 *Htr1a* (−/−)			
1.2		C57/Bl6 *Htr1a* (+/+)	0.2 mg/kg		
1.3		C57/Bl6 *Htr1a* (−/+)			
no effect		C57/Bl6 *Htr1a* (−/−)			
5	45	rats HDS *	0.5 mg/kg	s.c.	[45]
1.2		rats RDS *			
0.7		rats LDS *			
2.5	30	rats HDS *	20 µg	i.c.v.	
no effect		rats LDS *			
1.27	90	Sprague-Dawley rats	500 µg/kg	s.c.	[71]
1.15			0.5 µg	into the hypothalamus	
1.6	30	Sprague-Dawley rats	0.05 mg/kg	s.c.	[72]
3.5			0.1 mg/kg		
2	30	Sprague-Dawley rats	0.1 mg/kg	i.p.	[50]
3			0.3 mg/kg		
3.5			1 mg/kg		
2.1	30	♀♂mice 129rSv WT	0.2 mg/kg	s.c.	[73]
3.1	30	♀♂ *Htr1b* KO mice	0.6 mg/kg		
2.5	30	Sprague-Dawley rats	0.1 mg/kg	s.c.	[74]
2	30	Sprague-Dawley rats	0.3 mg/kg	i.p.	[75]
0.5	30	C57BL/6J mice	0.3 mg/kg	i.p.	
1.2			0.6 mg/kg		
1.5			0.1 mg/kg		
1.58	30	non aggressive rats	0.5 mg/kg	i.p.	[47]
no effect		aggressive rats			
no effect	30	aggressive rats	0.5 mg/kg	i.p.	[76]
2.2		non-aggressive rats			
0.75	20	Tg8 mice	2 mg/kg	i.p.	
2		C3H mice			
1.75	20	CBA/Lac mice	0.5 mg/kg	i.p.	[48]
2.25			1 mg/kg		
no effect	20	AKR mice	0.5 mg/kg1 mg/kg		
1.4	30	Sprague-Dawley rats	0.2 mg/kg	i.p.	[77]
1.57	10–20	CBA/Lac mice	1 mg/kg	i.n.	[78]
1.1				i.p.	
1.9				s.c.	
0.71		DBA/2 mice		i.n.	
1.37				i.p.	
1.44				s.c.	
1.8		PT mice		i.n.	
1.91				i.p.	
0.98		C57Bl mice		i.n.	
0.67				i.p.	
2.62				s.c.	
1.2	20	CBA/Lac mice	1 mg/kg	i.p.	[79]
1		C57/Bl6 mice			
1		DBA/2 mice			
2.2		BALB/C mice			
0.5		AKR mice			
0.5		ICR mice			
3	20	1A-High mice	0.5 mg/kg	i.p.	[51]
1	10	1A-Low mice			
2.25	20	CBA mice	1 mg/kg	i.p.	[49]
2.5		ASC mice			
0.4ns		AKR mice			
1.25		AKR.CBA-D13Mit76C mice			
2.5	30	CBA/Lac mice	40 nM	i.c.v.	[80]
1.2	20	CBA/Lac mice	1 mg/kg	i.p.	[30]
0.9		C57/Bl6 mice			
0.9		DBA/2 mice			
2.2		BALB/C mice			
0.5ns		AKR mice			
0.5ns		ICR mice			
1.2		C3H mice			
2.6		Asn mice			
3	20	WT mice	0.5 mg/kg	i.p.	[52]
1.3		KO *Htr1a* in DRN			
1.5	20	♂ WT mice	0.75 mg/kg	i.p.	[53]
1.75		♀ WT mice			
no effect		*1AcKO* mice ♂♀			

Note: 8-OH-DPAT—(±)-8-Hydroxy-2-(dipropylamino)tetralin; i.c.v.—intracerebroventricular; s.c.—subcutaneous; i.p.—intraperitoneal; i.v.—intravenous; i.n.—intranasal; DRN—dorsal raphe nucleus; MRN—median raphe nucleus; ns—not significant. *—Rats with high (HDS), low (LDS), and random (RDS) hypothermic responses to the 5-HT_1A_ receptor agonist 8-OH-DPAT. If it is not indicated specially, only the male were used in the experiments.

**Table 2 biomolecules-11-01914-t002:** Results of investigations of localization of 5-HT_1A_ receptors mediating the 8-OH-DPAT-induced hypothermia.

Conclusion	Argumentation	Object	References
Pre-	Hypothermic effect of 8-OH-DPAT was attenuated when 5-HT was depleted by repeated i.p. administration of pCPA or by injection of 5,7-DHT into the third ventricle	C57/B16/0la mice	[54]
Post-	After pretreatment by pCPA the minimal dose of 8-OH-DPAT (0.03 mg/kg s.c.), ineffective earlier, deveined effective. It may be the result of receptor sensitization that appeared after pCPA reducing of the whole brain 5-HT level.	Sprague-Dawley rats	[55]
Pre-	Hypothermic effect of 8-OH-DPAT was attenuated when 5-HT was depleted by repeated i.p. administration of pCPA or by injection of 5,7-DHT into the third ventricle	Sprague-Dawley rat	[58]
Post-	pCPA tended to increase the hypothermic effect of 8-OH-DPAT (−3 °C in comparing with −2,5 °C)	Sprague-Dawley rats	[59]
Post-	pCPA tended to increase the hypothermic effect of 8-OH-DPAT	ddY mice	[85]
Pre-	Lesioning of central 5-HT neurones by 5,7-DHT abolished the hypothermic response to 8-OH-DPAT. Depletion of brain 5-HT levels with p-CPA markedly attenuated the hypothermic response.	Female albino Tuck (T/O strain) mice	[64]
Post-	Lesioning of central 5-HT neurones by 5,7-DHT had no effect on the hypothermic responses to lower doses of 8-OH-DPAT. pCPA significantly increased hypothermic response.	Sprague-Dawley rats	[64]
Pre- Post-(?)	Injection of 8-OH-DPAT directly into the DRN evoked a clear hypothermic effect. Injection of 8-OH-DPAT directly into the MRN produced no hypothermic effect.	Wistar rat	[9]
?	pCPA pretreatment reduced the 5-HT level in cortex and hippocampus, but did not affect the hypothermia	Swiss-Webster mice	[65]
Pre-	Destruction of 5-HT neurones with the neurotoxin 5,7-DHT abolished the hypothermic response to 8-OH-DPAT	C57/Bl/601a mice	[66]
Post-	Dose–response curve shows that i.c.v. administration of 8-OH-DPAT was more potent than direct injection to the DRN to causing hypothermia.	Sprague-Dawley rats	[81]
?	Correlation between the severity of hypothermic response to 8-OH-DPAT and radio ligand 5-HT_1A_ binding sites in the brain regions associated with thermoregulation was not found	rats HDS * rats LDS *	[45]
Post-	Buspirone produced the same significant increase in prolactin and growth hormone (effects mediated by post- 5HT_1A_ receptors) in the tryptophan-depleted state as in the control condition. The degree of hypothermia produced by buspirone was not significantly different in the two experimental conditions.	healthy volunteers humen	[86]
Pre-	*Htr1a* auto-KO mice displayed no detectable body temperature decrease in response to 8-OH-DPAT	Genetically modified mice	[53,87]
Pre-	*Htr1a* hetero-KO mice displayed a full hypothermic response to 8-OH-DPAT	Genetically modified mice	[87]

Note: The table does not include the data of O’Connell et al. [88], as well as a number of other authors, despite the promising titles of their works. O’Connell et al. (1992) [88] used in their study the substance BMY7378, which, as was showed later, was primarily an antagonist of α-adrenergic receptors (IUPHAR / BPS Guide to Pharmacology) [89]. So it is possible that the observed effects of this compound were the results of its action on adrenergic, but not 5-HT receptors. 8-OH-DPAT—(±)-8-Hydroxy-2-(dipropylamino)tetralin; pCPA—para-chlorophenylalanine; 5,7-DHT—5,7-dihydroxytryptamine; i.p.—intraperitoneal; s.c.—subcutaneous; i.c.v.—intracerebroventricular; DRN—dorsal raphe nucleus; MRN—median raphe nucleus; Post-(?)—role of post-synaptic receptors is not completely excluded; ?—no defined answer; *—Rats with high (HDS) and low (LDS) hypothermic responses to the 5-HT1A receptor agonist 8-OH-DPAT.

**Table 3 biomolecules-11-01914-t003:** Effect of 5-HT_1A_ receptor activation by its selective agonist 8-OH-DPAT on thermoregulatory responses in warm-blooded organisms.

Change in Thermoregulatory Response	Change in Registered Parameter	Ambient Temperature Conditions	Object	Dose and Route of Administration	References
↓General metabolism	↓O_2_ consumption	Room temperature	rat	500 µg/kg s.c.	[71]
			rat	0.5 µg into the hypothalamus	[71]
			mice	40 nM left lateral ventricle	[80]
	↓excretion of CO_2_ increased via leptin injection		rat	10 mM 60 nL raphe’ pallidus	[113]
	↓O_2_ consumption	Cooling	rat	4 mg/mL 1 µL Nucleus raphe’ magnus	[120]
	↓excretion of CO_2_		rat	10 mM 60 nL Rostral raphe’ pallidus	[115]
			rat	100 µg/500 µL i.v.	[114]
			rat	10 mM 60 nL Rostral raphe’ pallidus	[114]
↑Heat loss	↑skin temperature	Room temperature	rat	500 µg/kg s.c.	[71]
			rat	0.5 µg into the hypothalamus	[71]
			rat	0.6 µmol/kg s.c.	[123]
	↑peripheral blood flow		rabbit	0.1 mg/kg i.v.	[98]
	↑peripheral blood flow	Cooling	rabbit	0.1 mg/kg i.v.	[99]
			rabbit	3–5 nmol rostral medullary raphe’	[100]
			rat	0.5 mg/kg, s.c.	[101]
	↑skin temperature		piglet	dialysis 30 mM 30 min medullary raphe’	[122]
			rat	0.6 µmol/kg s.c.	[123]
↓Heat production	↓BAT temperature increased via leptin injection	Room temperature	rat	10 mM 60 nL raphe’ pallidus	[113]
	↓shivering, i.e., muscle electrical activity	Cooling	rat	4 mg/mL 1 µL Nucleus raphe’ magnus	[120]
			piglet	dialysis 30 mM 30 min medullary raphe’	[122]
			piglet	dialysis 30 mM 30 min paragigantocellularis lateralis	[121]
	↓BAT temperature		rat	10 mM 60 nL Rostral raphe’ pallidus	[115]
			rat	0.5 mg/kg, s.c.	[101]
			rat	100 µg/500 µL i.v.	[114]
			rat	10 mM 60 nL Rostral raphe’ pallidus	[114]

Note: ↓—decrease of parameter; ↑—increase of parameter.

**Table 4 biomolecules-11-01914-t004:** Effect of 5-HT_2A/C_ agonists and antagonists on body temperature of warm-blooded animals.

Compound	Dose and Route of Administration	Temperature Effect (Time of Its Achievement)	Object	Ambient Temperature Conditions	References
**Effect of 5-HT_2A/C_ agonists**					
MK-212	0.03; 0.1; mg/kg i.p.	no effect (30′)	Sprague- Dawley rats	28–29.5 °C	[56]
	1.0 mg/kg i.p.	0.8 °C (15′)			
		0.83 °C (30′)			
	3.0 mg/kg i.p.	1.1 °C (30′)			
MK-212	2.5 mg/kg i.p	1.4 °C (30′)	Sprague- Dawley rats	29 °C	[62]
mCPP	1.25 mg/kg i.p.	0.5 °C (30′)	Wistar rats	25 °C	[158]
	2.5 mg/kg i.p.	0.75 °C (15′);			
		1.1 °C (30′);			
		1.1 °C (45′)			
	5 mg/kg i.p.	0.75 °C (15′);			
		1.1 °C (30′);			
		1.2 °C (45′);			
		1.3 °C (60′);			
		1.3 °C (90′)			
mCPP	0.5 mg/kg oral	0.48 °C (≈190′)	Health volunteers		[159]
	0.1 mg/kg i.v.	0.32 °C (≈57′)			
DOI	0.1 mg/kg i.p.	0.4 °C (30′)	Sprague- Dawley rats	29 °C	[160]
	0.3 mg/kg i.p	1.3 °C (30′)			
	1.0 mg/kg i.p.	1.8 °C (30′)			
DOI	3 mg/kg s.c.	1.5 °C (30′)	Sprague- Dawley rats	20 °C	[161]
DOI	0.8 mg/kg i.v.	0.9 °C (5′); 1.5 °C (10′); 2 °C (15′); 2.5 °C (20′); 3 °C (30′) Malignant hyperthermia	pigs		[162]
TFMPP	1 mg/kg i.p.	0.7 °C (30′); 0.6 °C (60′);	Wistar rats	28 °C	[163]
	5 mg/kg i.p.	0.9 °C (30′); 1.0 °C (60′); 0.9 °C (90′); 0.6 °C (120′);			
	10 mg/kg i.p.	1.0 °C (30′); 1.3 °C (60′); 1.4 °C (90′); 1.1 °C (120′);			
	20 mg/kg i.p.	0.8 °C (30′); 1.1 °C (60′); 1.3 °C (90′); 1.3 °C (120′);			
mCPP	1 mg/kg i.p.	0.6 °C (60′); −0.5 °C (90′)			
	5 mg/kg i.p.	0.6 °C (60′); 0.9 °C (90′)			
	10 mg/kg i.p.	1.1 °C (30′); 1.2 °C (60′);			
		1.1 °C (90′); 0.9 °C (120′);			
	20 mg/kg i.p.	1.3 °C (30′); 1.3 °C (60′); 1.2 °C (90′); 1.2 °C (120′);			
DOI	1 mg/kg i.p.	1.5 °C (60′)	Wistar rats	21 °C	[164]
DOI	2.5 mg/kg i.p.	1.5 °C (60′)	Wistar rats	22 °C	[165]
		0.6 °C (60′)	Fawn- Hooded rats		
mCPP	2.5 mg/kg i.p.	1 °C (30′)	Wistar rats		
		0.3 °C (30′)	Fawn- Hooded rats		
mCPP	2.5 mg/kg i.p.	1.2 °C (30′)	Wistar rats	21 °C	[166]
		0.4 °C (60′)	Fawn- Hooded rats		
	1.25 mg/kg i.p.	0.5 °C (30′)	Wistar rats		
		no effect	Fawn- Hooded rats		
DOI	2.5 mg/kg i.p.	2 °C (60′)	Wistar rats		[167]
mCPP		1.2 °C (30′)			
DOI	1 mg/kg s.c.	1.2 °C (30′)	Wistar rats	23 °C	[69]
DOI	Intrahypothalamic administration 0.2 µg	1.38 °C (45′)	Sprague- Dawley rats	*room temperature*	[71]
DOI	0.025; 0.1 mg/kg s.c.	no effect (20′)	Sprague- Dawley rats	21 °C	[168]
	0.4; 1.6 mg/kg s.c.	1.0 °C (20′)			
DOI	5 µg/kg i.v.	0.25 °C (20′–60′)	rabbits	23–25 °C	[169]
	50 µg/kg i.v.	1.0 °C (40′–60′)			
	100 µg/kg i.v.	>2 °C (90′–120′)			
	100 µg/kg s.c.	1.0 °C (90′)	Sprague- Dawley rats	27 °C	
YM348	0.3 mg/kg per os	0.9 °C (60′)	Zucker rats		[170]
	1 mg/kg per os	1.1 °C (60′)			
	3 mg/kg per os	1.3 °C (60′)			
DOI	0.1 mg/kg s.c.	1.4 °C (30′)	Sprague- Dawley rats	25–28 °C	[101]
DOI	0.05 mg/kg s.c.	0.29 °C ns	Sprague- Dawley rats	22 °C	[171]
	0.1 mg/kg s.c;.	0.64 °C (30′; 45′; 60′)			
	0.5 mg/kg s.c.	0.79 °C (15′; 30′; 45′; 60′; 90′)			
	0.05 mg/kg s.c.	0.5 °C ns		32 °C	
	0.1 mg/kg s.c.	0.5 °C (15′); 1.7 °C (75′)			
	0.5 mg/kg s.c.	1.2 °C (15′); 2.7 °C (75′)			
	0.1 mg/kg s.c.	0.1°C ns		12 °C	
	0.1 mg/kg s.c.	0.99 °C (30′; 45′; 60′)		27 °C	
25B-NBOMe	0.25 mg/kg i.p.	no effect	Sprague- Dawley rats	23 °C	[172]
	0.25 mg/kg i.p.	0.8 °C (30′); 1 °C (60′); 1.2 °C (90′); 0.9 °C (120′)		29 °C	
**Effect of 5-HT_2A/C_ antagonists**					
pirenpirone	0.01; 0.03; 0.1; 0.3 mg/kg i.p.	−0.15 °C ?s; −0.5 °C; −0.85 °C; −0.9 °C (<60′)	Sprague- Dawley rats	28–29.5 °C	[56]
ketanserin	0.1; 0.3; 1; 3 mg/kg i.p.	−0.2 °C ?s; −0.5 °C; −0.7 °C; −1.2 °C (<60′)			
ritanserin	0.63 mg/kg i.p.	no effect	Wistar rats	25 °C	[158]
ritanserin	0.5; 1; 2 mg/kg i.p	no effect alone, but dose-dependently reduces hyperthermic effect of mCPP and TFMPP	Wistar rats	28 °C	[163]
ketanserin	0.6; 1.25; 2.5 mg/kg i.p				
LY53857	1 mg/kg i.p.	−0.55 °C (90′)	Wistar rats	21 °C	[164]
ketanserin	1 mg/kg i.p.	no effect			
ritanserin	1 mg/kg s.c.	no effect (30′)	Sprague- Dawley rats	21 °C	[168]
amperozide	2 mg/kg s.c.	no effect (30′)			
ketanserin	5 mg/kg i.p	no effect	Sprague- Dawley rats	22 °C	[171]
		no effect alone, but blocks DOI		32 °C	
ketanserin	0.2 µg into preoptic anterior hypothalamus	0.58 °C (180′)	rabbit	2 °C	[173]
		0.54 °C (180′)		22 °C	
		0.57 °C (180′)		32°C	
	0.4 µg into preoptic anterior hypothalamus	1.13 °C (180′)		2 °C	
		1.0 °C (180′) *		22 °C	
		1.09 °C (180′)		32 °C	
	2 µg into preoptic anterior hypothalamus	1.42 °C (180′)		2 °C	
		1.37 °C (180′)		22 °C	
		1.43 °C (180′)		32°C	

Note: MK-112—2-chloro-6-piperazin-1-ylpyrazine; mCPP—1-(m-chlorophenyl)piperazine meta chlorophenyl piperazine; DOI—1-(4-iodo-2,5-dimethoxyphenyl)propan-2-amine; TFMPP—1-[3-(Trifluoromethyl)phenyl]piperazine); YM348—(2S)-1-(7-ethylfuro[2,3-g]indazol-1-yl)propan-2-amine; 25B-NBOMe—2-(4-Bromo-2,5-dimethoxyphenyl)-N-(2-methoxybenzyl)ethanamine; LY53857—3-hydroxybutan-2-yl(2R,4R,7R)-6-methyl-11-(propan-2-yl)-6,11-diazatetracyclo[7.6.1.0^{2,7}.0^{12,16}]hexadeca-1(16),9,12,14-tetraene-4-carboxylate; *—the value is taken from the text of the article, not from the table; ns—non significant; ?s—the authors did not indicate the significance.

**Table 5 biomolecules-11-01914-t005:** Effect of 5-HT_2A/C_ agonists and antagonists on body temperature of mice.

Compound	Dose and route of Administration	Temperature Effect (Time of Its Achievement)	Object	Ambient Temperature Conditions	References
**Effect of 5-HT_2A/C_ agonists**					
TFMPP	1 mg/kg s.c.	no effect alone(10′), but reduces the effect of 8-OH-DPAT	ICR	21 °C	[63]
mCPP					
MK-212					
DOI	0.1 mg/kg s.c.				
MK-212					
DOI	0.5, 1.0, 2.5 mg/kg i.p.	no effect (15′, 30′, 45′, 60′, 90′, 120′)	ICR	24 °C	[174]
	5 mg/kg i.p.	−1 °C (30′, 45′, 60′, 90′)			
DOI	1 mg/kg i.p; 5 mg/kg i.p.	no effect	Ddy	24 °C	[175]
DV-7028	10 mg/kg per os				
TCB-2	0.1; 0.5; 1.0; 2.5; mg/kg i.p.	no effect (15′, 30′, 45′)	C57Bl6J	plexiglass containers 20 °C	[176]
	5.0 mg/kg i.p.	−4 °C ( ‘)			
DOI	0.1; 0.5; 1.0; 2.5; 5.0 mg/kg i.p.	no effect (15′, 30′, 45′)			
DOI	1 mg/kg i.p.	no effect (each 5′ over 24 h)	C57BL/6J	22.5–23.5 °C	[177]
25CN-NBOH	1.5 mg/kg, s.c.	no effect	Mice Mix genetic background	Temperature of heat-pad = 37 °C	[178]
		1 °C (15′)		Temperature of heat-pad = 41 °C	
**Effect of 5-HT_2A/C_ antagonists**					
ritanserin	1 mg/kg per os 10 mg/kg per os	no effect	Ddy	24 °C	[175]
ketanserin	1 mg/kg per os	no effect			
	10 mg/kg per os	−2.2 °C (60′)			
MDL 11,939	(1.0 mg/kg), i.p.	no effect (30′)	C57Bl6J		[176]
ketanserin	1.0 mg/kg i.p. (180 µM).	−2 °C	CBA/Lac		[179]
	2.0 mg/kg i.p. (360 µM)	−2.3 °C			
	20 nM i.c.v.	−2.3 °C			
	40 nM i.c.v.	−3.5 °C			
ketanserin	1 mg/kg i.p.	no effect (each 5′ over 24 h)	C57BL/6J	22.5–23.5 °C	[177]
	40 nmol i.c.v.	no effect (60′)			
ketanserin	1 mg/kg i.p.	no effect (60′)	AKR/J		[180]
			CBA/Lac		
			AKR.CBA- D13Mit76		

Note: DV-7028—3-[2-[4-(4-Fluorobenzoyl)-1-piperidinyl]ethyl]-6,7,8,9-tetrahydro-2*H*-pyrido[1,2-*a*]-1,3,5-triazine-2,4(3*H*)-dione hydrochloride; TCB-2—(4-Bromo-3,6-dimethoxybenzocyclobuten-1-yl)methylamine hydrobromide; 25CN-NBOH—N-(2-hydroxybenzyl)-2,5-dimethoxy-4-cyanophenylethylamine; MDL 11,939—α-Phenyl-1-(2-phenylethyl-4-piperidinemethanol; ( ‘)—the authors did not indicate the exact time for the effect achievement.

**Table 6 biomolecules-11-01914-t006:** Effect of 5-HT_2_ receptor activation by their selective agonists on thermoregulatory responses in warm-blooded organisms.

Change in Thermoregulatory Response	Change in Registered Parameter	Ambient Temperature Conditions	Object	Ligand (Dose and Route of Administration)	Reference
↑General metabolism	↑O_2_ consumption	Room temperature	Sprague- Dawley rat	DOI (0.2 µg into the hypothalamus)	[71]
	↑O_2_ consumption; CO_2_ excretion		Zucker fa/fa rats	YM348 (0.3; 1; 3 mg/kg per os)	[170]
↑ Heat production	↑BAT temperature	25–28 °C	Sprague- Dawley rat	DOI (0.1 mg/kg s.c)	[101]
	↑BAT temperature	29 °C	Sprague- Dawley rat	25B-NBOMe (0.25 mg/kg i.p.)	[172]
↓Heat loss	↓skin temperature	Room temperature	Sprague- Dawley rat	DOI (0.2 µg into the hypothalamus)	[71]
	↓peripheral blood flow	23–25 °C	Rabbits	DOI (5; 50; 100 µg/kg, i.v.)	[169]
		27 °C	Sprague- Dawley rat	DOI (100 µg/kg s.c.)	
	↓peripheral blood flow		rabbits	DOI (50 µg/kg, i.v.)	[183]
	↓peripheral blood flow	25–28 °C	Sprague- Dawley rat	DOI (0.1 mg/kg s.c)	[101]
	↓skin temperature	29 °C	Sprague- Dawley rat	25B-NBOMe (0.25 mg/kg i.p.)	[172]

Notes: ↓—decrease of parameter; ↑—increase of parameter.

**Table 7 biomolecules-11-01914-t007:** Effect of 5-HT_3_ agonists and antagonists on body temperature in warm-blooded animals.

Compound	Dose (Route of Administration)	Maximal Effect (°C)	Time of Its Achievements	Object	References
mCPBG	0.1 mg/kg (i.p.)	no effect		Wistar rats	[218]
	1 mg/kg (i.p.)	0.5	≈30′		
	10 mg/kg (i.p.)	0.6	≈30′		
MDL72222	0.1; 1; 10 mg/kg (i.p.)	no effect			
ondansetron	0.1; 1 mg/kg(i.p.)	no effect			
2-Me-5-HT	5 µg (i.c.v.)	≈−0.62	45′	Sprague- Dawley rats	[219]
	10 µg (i.c.v.)	−1.1	45′		
	20 µg (i.c.v.)	−1.5	45′		
	1–5 mg/kg (i.p.)	no effect	During 2 h		
ondansetron	5, 10, 20 µg (i.c.v.)	no effect	During 150′		
	50, 100, 300 mg/kg (i.p.)	no effect			
tropisetron	50, 100, 300 mg/kg (i.p.)	no effect			
	5, 10, 20 µg (i.c.v.)	no effect			
2-Me-5-HT	4 mg/kg (i.p.)	no effect	During 6 h	Sprague- Dawley rats	[220]
mCPBG	160 nmol (i.c.v.)	−4	30′	AKR/J mice	[79]
	80 nmol (i.c.v.)	−2			
	40 nmol (i.c.v.)	−2			
	20 nmol (i.c.v.)	no effect			
	0.5; 1; 5; 10 mg/kg (i.p.)	no effect			
mCPBG	40 nmol (i.c.v.)	−4	30′	DBA/2J mice	
		−5.5		CBA/Lac mice	
		−2.6		AKR/J mice	
		−5.1		C57Bl/6 mice	
		−3.1		BALB/c mice	
		−3.2		ICR mice	
mCPBG	40 nmol (i.c.v.)	−2.3	30′	CBA/Lac mice	[80]
mCPBG	40 nmol (i.c.v.)	−5	30′	C57Bl/6 mice	[30]
		−5		C3H mice	
		−4		DBA mice	
		−5.9		Asn mice	
		−3		BALB mice	
		−5.1		CBA mice	
		−3.2		ICR mice	
		−2.5		AKR mice	

Note: mCPBG—m-chlorophenylbiguanide; MDL72222—bemesetron; 2-Me-5-HT—2-methyl-5-hydroxytryptamine maleate.

**Table 8 biomolecules-11-01914-t008:** Changes in body temperature of mice lacking the *Htr7* gene (*Htr7* KO) and control (WT) ones after intraperitoneal injection of compounds inducing hypothermia.

Hypothermia-Inducing Compound	Dosa (mg/kg)	ΔT Body (Time of Effect)	Object	Reference
5-HT	5	−3.1 °C (30′)	WT	[242]
		−2 °C (60′)		
		no change	*Htr7* KO	
5-CT	0.5	−2 °C (30′)	WT	
	3	−4.5 °C (120′)		
	0.5	no change	*Htr7* KO	
	3	−1.2 °C (120′)		
5-CT	0.1	−1.2 °C ns	WT	[241]
	0.3	−2.8 °C		
	1	−3.9 °C		
	0.1; 0.3; 1	no change	*Htr7* KO *	
8-OH-DPAT	0.3	−0.7 °C (30′)	WT	[75]
	0.6	−1.2 °C (30′)		
	1	−1.5 °C (30′)		
	0.3	no change	*Htr7* KO	
	0.6	no change		
	1	−1.1 °C (30′)		

Note: 5-CT—5-carboxamidotryptamine maleate; *—background is 50%129SvEv/50% C57Bl/6J; for the others—C57BL/6J; ns—result is not significant.

**Table 9 biomolecules-11-01914-t009:** Impact of pretreatment with 5-HT_7_ and 5-HT_1A_ antagonists on hypothermic effects of 5-CT and 8-OH-DPAT in warm-blooded animals.

Pretreatment			Main Hypothermic Effect			Joint Effect of Two Compaunds ΔT Body	Object	References
Pretreatment Compound	Dose (Route of Administration)	Time Interval	Compound	Dose (Route of Administration)	ΔT body (Time of Effect)			
			5-CT	0.3 mg/kg (i.p.)	−1.8 °C (75′)		Guinea pig	[239]
SB-269970-A	3 mg/kg (i.p.)	60′	5-CT			−0.4 °C		
			5-CT	0.3 mg/kg (i.p.)	−1.9 °C		Guinea pig	[240]
SB-656104-A	3 mg/kg (i.p.)	60′	5-CT			−0.7 °C		
			5-CT	0.3 mg/kg (i.p.)	≈−1.8 °C		Swiss Webster mice	[241]
WAY100635	0.1; 0.3; 1 mg/kg (i.p.)	30′	5-CT	0.3 mg/kg (i.p.)		≈−1.4 °C		
SB 258719	5; 10 mg/kg (i.p.)	30′	5-CT	0.3 mg/kg (i.p.)		≈−1.7 °C		
	20 mg/kg (i.p.)					≈−0.2 °C		
SB-269970	1 mg/kg (i.p.)	30′	5-CT	0.3 mg/kg (i.p.)		−1.2 °C		
	3 mg/kg (i.p.)					−0.6 °C		
	10; 30 mg/kg (i.p.)					−0.4 °C		
			8-OH-DPAT	0.3 mg/kg (i.p.)	-2°C (30′)		Sprague–Dawley rats	[75]
WAY-100135	1 mg/kg (i.p.)	20′	8-OH-DPAT	0.3 mg/kg (i.p.)		−1.6 °C		
	10 mg/kg (i.p.)					−0.5 °C		
SB-269970	0.3 mg/kg (i.p.)	20′				−1.2 °C		
	3 mg/kg (i.p.)					−1.1 °C		
DR-4004	3 mg/kg (i.p.)	20′				−1.2 °C		
	10 mg/kg (i.p.)					−1.1 °C		
			8-OH-DPAT	0.3 mg/kg (i.p.)	−0.7 °C (30′)		C57Bl6+/+	
				0.6 mg/kg (i.p.)	−1.2 °C (30′)			
				1 mg/kg (i.p.)	−1.5 °C (30′)			
				0.3 mg/kg (i.p.)	no change		C57Bl6−/−	
				0.6 mg/kg (i.p.)	no change			
				1 mg/kg (i.p.)	−1.1 °C (30′)			
WAY-100135	10 mg/kg (i.p.)	30′	8-OH-DPAT	0.3 mg/kg (i.p.); 0.6mg/kg (i.p.); 1 mg/kg (i.p.)		no change	C57Bl6+/+	
						no change	C57Bl6−/−	
SB-269970	10 mg/kg (i.p.)	30′	8-OH-DPAT	0.3 mg/kg (i.p.)		no change	C57Bl6+/+	
				0.6 mg/kg (i.p.)		−0.7 °C (30′)		
				1 mg/kg (i.p.)		−1.9 °C (30′)		
				0.3 mg/kg (i.p.)		no change	C57Bl6−/−	
				0.6 mg/kg (i.p.)		−0.6 °C (30′)		
				1 mg/kg (i.p.)		−1.2 °C (30′)		
			8-OH-DPAT	0.1 mg/kg (s.c.)	−3.4 °C (30′)		Sprague–Dawley rats	[245]
WAY-100635	0.005 mg/kg, s.c.)		8-OH-DPAT			−2 °C		
SB-269970	0.1 mg/kg (i.p.)		8-OH-DPAT			−3.4 °C		
	0.5 mg/kg (i.p.)					−2.7 °C		
	1 mg/kg (i.p.)					−2.3 °C		
WAY-100635 (0.005 mg/kg, s.c.) + SB-269970(1 mg/kg, i.p.)			8-OH-DPAT			−1 °C		

Note: All compound used for pretreatment did not change animal’s body temperature when injected alone or together; ΔT body—change in body temperature; SB-269970-A—3-[(2R)-2-[2-(4-methylpiperidin-1-yl)ethyl]pyrrolidine-1-sulfonyl]phenol; SB-656104-A—6-[(2R)-2-[2-[4-(4-chlorophenoxy)piperidin-1-yl]ethyl]pyrrolidin-1-yl]sulfonyl-1H-indole; WAY100635—N-[2-[4-(2-methoxyphenyl)piperazin-1-yl]ethyl]-N-pyridin-2-ylcyclohexanecarboxamide; SB-258719—N,3-dimethyl-N-[(2R)-4-(4-methylpiperidin-1-yl)butan-2-yl]benzenesulfonamide; WAY-100135—(S)-N-tert-butyl-3-(4-(2-methoxyphenyl)piperazine-1-yl)-2-phenylpropanamide; DR-4004—2a-[4-(4-phenyl-3,6-dihydro-2H-pyridin-1-yl)butyl]-1,3,4,5-tetrahydrobenzo[cd]indol-2-one.

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
