# Peer review of "5-HT Receptors and Temperature Homeostasis"

_biomolecules, 2021, doi:10.3390/biom11121914_

Round 1

Reviewer 1 Report

Irina Petrovna Voronova proposes a review article on the control of serotonin (5-HT) upon body temperature in warm-blooded animals. The author focused on some 5-HT receptors that have been the object of several studies, namely 5-HT1A, 5-HT2, 5-HT3 and 5-HT7 receptor subtypes. 5-HT1A, 5-HT3 and 5-HT7 receptors may decrease body temperature whereas 5-HT2 receptors tend to increase it. The mechanisms behind the thermoregulatory action of each receptor are complex in terms of engagement of specific brain and peripheral tissues and pharmacology. The author tends toward the orchestration of all these influences on this fundamental, metabolic process without being able to materialize it because the data are still lacking. Overall, it could be an excellent review article with appropriate references, appropriate tables, some interesting elements of discussion, a good manipulation of concepts in thermoregulation. The main concern is the text including the style, the English, the typos, etc.

  1. The text. The style is not always appropriate which make me thinking that the author has poor experience in writing articles and more precisely review articles. The paroxysmal illustration can be found line 1043: “(it is not a typo: under the influence of E57421 the temperature drop in WT was less than that in Htr7K/O!).” There is also line 882: “Let me remind you….”, line 566: “This is not speculative conclusion”, line 1201: “Note, that the same processes occur when serotonin receptors of 1A and 3 types are activated!” Etc. The text is full of inappropriate words, or phrasing. The author also uses double negation such as “it did not give an unambiguous answer”. Thus, I recommend that the text can be read and corrected by a researcher with dense research background, and to give the corrected version for English corrections. At present, there are some parts that were not that clear to me and it is simply due to the combination of lay style and English matters.
  2. The tables are great but they have to be ameliorated. First the correspondence of the lines is not always clear and it looks like there are overlaps in some cases that should not be there. I would give more space between studies. Second, I don’t want to see “mkg” which means nothing. I guess it has something to do with µmol, and/or µg. The author has to correctly report the dose and the unit. For instance, table 1 8-OH-DPAT has been administered at 500 mkg/kg… The same in other tables and of course, it has to be corrected. It has to be corrected in the text too. The route of administration
  3. Typos are all over the text, tables, and even the figure 1: it could be pretriment, para-chlophinylalanine, spregue-dawley, rout, Melller, periferal, 20 nmo, termogenesis………..
  4. Consistency in reporting 5-HT receptors is not even met. Sometimes there is “serotonin 5-HT receptors”, sometimes “5-HT receptors”, and sometimes “5-HT”. Serotonin is also abbreviated, or not. It is pointless to indicate serotonin 5-HT1A receptor for instance.
  5. The introduction of the receptors is more or less correct although it could be reduced. It does not help understanding the action of pharmacological compounds and it is sometimes beyond the scope of the review. For example, the paragraph starting line 429 is outside the scope. Moreover, the 5-HT2C mRNA does not have 4 editing sites as it can be read in the review. Appropriate references and information have to be reported.
  6. In the footnote legend of table 2, the author reports: “It is difficult to imagine that the same compound can activate receptors in one region and block them in another.” As a general comment, the day the author reaches the toe of Mark J Millan in terms of pharmacology of monoaminergic systems and receptors, she will be able to address this kind of comments. What is annoying in the comment of the author is that it reveals that the knowledge in pharmacology of the author looks fragile. Yet, the author correctly reports that transduction systems of 5-HT receptors are complex and multiple but the notion of “biased agonism” seems to be completely absent. It could have been the object of some comments as regards the effects of ligands (notably some 5-HT antagonists) on body temperature.
  7. The authors did not evoke the resulting effect of SSRI (which is probably in her mind as I see how she manages physiology of 5-HT system) or MDMA administration on body temperature.

Thus, to summarize, I agree with the need to have this review published, and I do believe it is a fundamental step in the understanding of the role of the 5-HT system in thermoregulatory processes. It was recalling me my studies on body temperature, trying to follow the Goodwin et al data, notably the desensitization of presynaptic 5-HT1A receptors and the corresponding loss of effect of DPAT. But the review cannot be published as it is.

Author Response

Dear Reviewer!

I thank You for the sincere interest to the article and useful comments.

  1. I thought that the style of my review could be more free than the style of an original article, that it should be written with an intriguing introduction, with unusual facts in the "general part", with a "lively" style (as a lecture).

I removed all sentences with emotions, left only the "intriguing introduction".

According to you recommendation the text were read and corrected by the English language specialist again.

  1. I changed “mkg” by “µg”, tried to remove the typos, returned the initial (readable) form for the tables and would like to hope that the tables will not change again during the submission.
  2. I verified the text and tried to remove the typos.
  3. I verified the text and eliminated the diversity in the receptors name.

5.1. I agree with you and removed the second part of the paragraph that in the first version starting at the line 429. It really is outside the scope of the review.

The 1st sentence is one of the general characteristic of the 5-HT2 receptor genes. I modified it and, I think, it can be leave.

5.2 I am very grateful for your note concerning RNA edition. I corrected this mistake and recompleted the list of references. Thank you.

  1. I remove the phrase from the footnote of the Table2. I completely agree with you, dear Reviewer, that it is my personal opinion.

But I hope that you let me give an example in my justification. This is an example of “different action of agonist” (8-OH-DPAT) on pre- and post- synaptic receptors (described by Saudou & Hen in their review of 1994, pp 334-336). 5-HT1A receptor is negatively coupled with adenilate cyclase (classic event), but in hippocampus increase in cAMP level was observed in response to 5-HT1A agonist. It was unexpected and surprising. Later 5-HT7 receptor that positively coupled with adenilate cyclase and had affinity to 8-OH-DPAT was found (Saudou, Hen, 1994; ref # 18).

  1. Your remark is quite equitable. I tried to avoid the works where authors used compounds with multiple targets. Since SSIRs increase the level of 5-HT in synaptic cleft, their effects on body temperature are the results of activation of all 5-HT receptors. And it is quite difficult to identify the share of each receptor in these processes. In the article I cite only 1 work where SSIR was used (â„– 254). The authors injected this compound into very local brain structure by dialysis.

As for MDMA – its effect on the brain is even more complex. It acts not only on 5-HT, but also on the others brain mediatory systems.

Reviewer 2 Report

The review on the effect of serotonin reseptors on body temperature in warm-blooded animals has undoubtedly been prepared very carefully. The work contains over 250 references, which proves that the author is well prepared for the topic.

The work is prepared carefully, but a small punctuation correction is still needed. The numbering next to the receptor name should be subscript.  In my opinion, the abstract should contain little information about the conclusions drawn by the author after studying the literature.

The article is written by one author, however, there are terms: „So, we have analyzed….”, „we also observed a significant”, „We would like to emphasize that” this should be changed, maybe the impersonal form would be the best.

There is also no reference to other similar works on this subject, e.g.

- Takayuki Ishiwata, Role of serotonergic system in thermoregulation in rats. J Phys Fitness Sports Med, 3(4): 445-450 (2014) DOI: 10.7600/jpfsm.3.445

- Rausch J.L. et al. Depressed Patients Have Higher Body Temperature: 5-HT Transporter Long Promoter Region Effects. Neuropsychobiology 2003;47:120–127 DOI: 10.1159/000070579

- Nagai M. The Role of Serotonergic System in Body Temperature Regulation. Physiol.  Res. 41: 65-69,  1992

- Kaplan K., Chronic central serotonin depletion attenuates ventilation and body temperature in young but not adult Tph2 knockout rats. J Appl Physiol 120: 1070–1081, 2016. doi:10.1152/japplphysiol.01015.2015

The article is very interesting and will surely be of interest to the readers of Biomolecules, so after completing and improving it, in my opinion, it should be published.

Author Response

Dear Reviewer!

I thank You for the sincere interest to the article and useful comments.

According to your recommendation:

  1. I corrected the receptors names;
  2. I rewrote the Abstract;
  3. I corrected the text and removed the sentences that you indicated.

I am very grateful to you for the list of references. These articles are very interesting. However the theme of my review is restricted by “Receptors”. So, I could use only 2 of them.

Round 2

Reviewer 1 Report

Irina Petrovna Voronova proposes a review article on the control of serotonin (5-HT) upon body temperature in warm-blooded animals. The author focused on some 5-HT receptors that have been the object of several studies, namely 5-HT1A, 5-HT2, 5-HT3 and 5-HT7 receptor subtypes. 5-HT1A, 5-HT3 and 5-HT7 receptors may decrease body temperature whereas 5-HT2 receptors tend to increase it. The mechanisms behind the thermoregulatory action of each receptor are complex in terms of engagement of specific brain and peripheral tissues and pharmacology. The author tends toward the orchestration of all these influences on this fundamental, metabolic process without being able to materialize it because the data are still lacking. Overall, it IS an excellent review article with appropriate references, appropriate tables, some interesting elements of discussion, a good manipulation of concepts in thermoregulation.

Now perfectly readeable, and to be cited.